# Enhanced and synergistic catalytic activation by photoexcitation driven S−scheme heterojunction hydrogel interface electric field

Aiwen Wang [1], Meng Du[1], Jiaxin Ni[1], Dongqing Liu[1], Yunhao Pan[1], Xiongying Liang[1], Dongmei Liu [1] ✉, Jun Ma[1], Jing Wang [2,3] ✉ & Wei Wang [1] ✉

The regulation of heterogeneous material properties to enhance the peroxymonosulfate (PMS) activation to degrade emerging organic pollutants remains a challenge. To solve this problem, we synthesize S−scheme heterojunction PBA/MoS$_2$@chitosan hydrogel to achieve photoexcitation synergistic PMS activation. The constructed heterojunction photoexcited carriers undergo redox conversion with PMS through S−scheme transfer pathway driven by the directional interface electric field. Multiple synergistic pathways greatly enhance the reactive oxygen species generation, leading to a significant increase in doxycycline degradation rate. Meanwhile, the 3D polymer chain spatial structure of chitosan hydrogel is conducive to rapid PMS capture and electron transport in advanced oxidation process, reducing the use of transition metal activator and limiting the leaching of metal ions. There is reason to believe that the synergistic activation of PMS by S−scheme heterojunction regulated by photoexcitation will provide a new perspective for future material design and research on enhancing heterologous catalysis oxidation process.

Peroxymonosulfate can be activated to produce reactive oxygen species (ROS) through methods such as light irradiation, ultrasound, heat, electricity and catalysts, achieving efficient degradation and mineralization of organic pollutants in wastewater[1]. Among them, transition metal catalysts exhibit good performance in PMS activation due to their diverse oxidation states, controllable operation, and low cost[2–4]. However, single-component metal catalysts often have insufficient catalytic activity due to the generation of high valence states during the reaction process. At the same time, the dispersion of powder catalysts in the reaction environment is difficult to recover and the leaching of metal ions is not conducive to their application. Therefore,

it is crucial to design new PMS activation materials with excellent catalytic performance while overcoming the shortcomings of transition metal-based catalysts[5].

Metal-organic frameworks (MOFs) have shown potential applications in PMS activation, but most MOFs only have a single metal and limited performance, and also face instability in aqueous solutions[6,7]. Prussian blue analogues (PBAs) have diversity and chemical stability in a wide pH range[8,9]. CoFePBA is a bimetallic MOF that exhibits synergistic effects between metals due to its multielectron and multinuclear configuration, greatly enhancing its catalytic activity and electron migration energy[10,11]. The

---

[1]State Key Laboratory of Urban Water Resource and Environment (SKLUWRE), School of Environment, Harbin Institute of Technology, Harbin 150090, P. R. China. [2]Institute of Environmental Engineering, ETH Zürich, Zürich 8093, Switzerland. [3]Laboratory for Advanced Analytical Technologies, Empa, Swiss Federal Laboratories for Materials Science and Technology, Dübendorf 8600, Switzerland. ✉e-mail: ldm819@126.com; jing.wang@ifu.baug.ethz.ch; wangweirs@hit.edu.cn

adsorption of PMS and intermediates, as well as the accompanying electron transfer, are key steps in the activation process, and the generation of ROS is also believed to be related to electrons. Therefore, the PMS activation can be accelerated by increasing the rate of electron generation to enhance ROS production[12–14]. Semiconductors with appropriate bandgap can provide photo-generated electrons under light irradiation, which has been proven to be an effective means of PMS activation[15]. These electrons can break the O−O bond in PMS molecules to produce $SO_4^{·-}$. However, the degradation efficiency of photocatalytic-activated PMS is still unsatisfactory to some extent due to the low efficiency of photo-induced carrier separation[16]. Specifically, the strong Coulomb interaction between individual semiconductor electrons and holes leads to a large binding energy of excitons, which hinders their separation, not to mention the insufficient light absorption in the visible light region[17]. Individual semiconductors can only provide weak redox driving forces due to insufficient light utilization, band positions and charge carrier mobility[18]. Fortunately, it has been proved that suitable heterojunction systems can be constructed to minimize exciton binding energy to effectively extract electrons[19,20]. The recently developed S−scheme heterojunction has an efficient charge transfer ability (Supplementary Fig. 1 and Supplementary Note 5)[21]. Meanwhile, electrons and holes generated by photoexcitation heterojunction accumulate at the more negative conduction band (CB) and positive valence band (VB) positions, respectively. Therefore, the S−scheme heterojunction enhances charge separation and redox capabilities. The interface interaction between the exposed metal and cyanide group (−CN) ligand in the PBA structure may promote light absorption ability, regulate band structure, and provide atomic level charge transfer channels[22–24]. Among the candidates for constructing heterojunction, molybdenum disulfide ($MoS_2$), a layered transition metal chalcogenide, exhibits great potential in PMS activation and catalysis[25]. In particular, the efficient electron transfer induced by the stacked atomic layers of S−Mo−S bonds is essential for its good catalytic properties. Therefore, the abundant active site ($Mo^{2+}→Mo^{6+}$) and suitable band gap of $MoS_2$ should theoretically be ideal catalysts for activating PMS and constructing heterojunction. However, how to establish an ideal charge transfer/separation interface to efficiently synergistically enhance the activation of PMS is still a challenge to be tackled.

In order to solve this problem, we design an S−scheme heterojunction PBA/$MoS_2$@chitosan hydrogel (CSH) for the heterogeneous catalysis oxidation process of synergistic activation of PMS under photoexcitation. Compared with the conventional PMS activation mechanism, we find that PBA/$MoS_2$@CSH not only can be used as a transition metal (CoFePBA) and a co−catalyst ($MoS_2$) to efficiently activate PMS through cycling between metal valence states, but also can activate PMS to remove doxycycline (DC) by photoexcited S−scheme heterostructure with efficient electron transfer enhanced by PBA/$MoS_2$ interface electric field. Unlike conventional powder catalysts, PBA/$MoS_2$@CSH is cross−linked into millimeter-sized hydrogel beads with 3D polymer chain closed network through natural polymer chitosan, which is conducive to rapid capture of PMS and electronic transmission in the advanced oxidation process, reducing the use of transition metal activator, and limiting the leaching of metal ions by its active −NH$_2$ and −OH groups[26,27]. Meanwhile, it can be easily separated from aqueous solutions, which provides possible convenience for practical applications. We have conducted in−depth studies on the mechanism of efficient electron transfer at photoexcitation S−scheme heterojunction interface electric field, revealing the synergistic effect of photogenerated carrier and PMS activation, and providing insights for the future development and application of novel PMS-activated functional materials in the field of water remediation.

## Results

### Morphological and catalytic characteristics of materials

The detailed synthesis route of PBA/$MoS_2$@CSH was shown in Supplementary Fig. 2. In short, PBA/$MoS_2$@CSH was prepared by acidizing chitosan with CoFePBA and $MoS_2$ powder. The −NH$_2$ in the chitosan polymer tended to form −NH$_3^+$ by protonation in an acid solution. Therefore, the entangled chitosan polymer chains expanded and dispersed due to repulsive forces, which provided many opportunities for the attachment of PBA/$MoS_2$ particles. In this process, PBA/$MoS_2$ was coordinated with a large number of active −NH$_2$ and −OH in the chitosan molecular chain to form cross-linking of adjacent chains[28]. Then, a simple pH inversion triggered the rapid gel of chitosan sol. The spontaneous entanglement, hydrogen bond and electrostatic force of the whole chitosan network stabilized all structural components in the gel, resulting in self−supporting PBA/$MoS_2$@CSH (Supplementary Fig. 3)[29]. Heterostructure composed of organic amorphous chitosan, PBA and $MoS_2$ layer can be observed in the Transmission Electron Microscopy (TEM) of PBA/$MoS_2$@CSH (Fig. 1a). The orthogonal lattice stripes at 0.25 and 0.63 nm in the high−resolution TEM (HR-TEM) image were attributed to the PBA (400) and $MoS_2$ (002) crystal planes[30,31]. The XRD pattern of Supplementary Fig. 4a showed that the relatively weak peak of the composite hybrid material PBA was due to the low content of PBA and low crystallinity of chitosan and $MoS_2$, so the effective crystal structure cannot be detected. Further element mapping and X−ray photoelectron spectroscopy (XPS) results (Fig. 1b and c) also confirmed that PBA and $MoS_2$ were bound in the CSH with uniform distribution of N, Co, Fe, Mo, and S elements. In addition, with the increased of $MoS_2$ content, the characteristic peaks at (002), (100) and (110) (JCPDS 75−1539) gradually increased (Supplementary Fig. 4b). However, the increase of $MoS_2$ content did not significantly improve the degradation of DC by PMS activation (Supplementary Fig. 5). Therefore, PBA/$MoS_2$(1:1)@CSH was the main research object for further characterization and exploration of catalytic performance considering catalyst cost. The Fourier transform infrared (FTIR) spectrum (Supplementary Fig. 6a) further confirmed the existence of Co−N≡C−Fe structure in the hydrogel sphere, and the main characteristic wavelengths were shown in Supplementary Table 1. Chitosan molecules contain three active groups, $C_2$−NH$_2$, $C_3$−OH, and $C_6$−OH, which participate in coordination and crosslinking reactions, resulting in a certain shift or superposition of peak values. The C≡N vibration peak of PBA in the wavenumber range of 2094 cm$^{-1}$ moved to 2048 cm$^{-1}$ after being wrapped in CSH. Because there were different coordination modes in the PBA structure, it had a certain number of open channels formed by unbounded nitroso and coordinated water molecules (Supplementary Fig. 6b inset)[10]. The electron paramagnetic response (EPR) spectrum of Supplementary Fig. 6b also provided evidence of vacancies in the prepared material. This was caused by distortion, possible bond isomerization of C≡N ligands, and multiple valence states of metal centers[32].

The difference in the catalytic performance of photoexcitation PBA/$MoS_2$@CSH activated PMS system was evaluated with DC as the target pollutant. It can be observed in Fig. 1d that the efficiency of PBA/$MoS_2$@CSH activating PMS was significantly improved when the photoexcitation was triggered. PBA/$MoS_2$@CSH was less effective (about 20%) in removing DC without the addition of PMS because they only rely on adsorption and less efficient photocatalysis. Figure 1e shown the synergistic structure−activity relationship between photoexcitation and PMS activation of the catalysts. The removal rate and catalytic rate constant k of PBA/$MoS_2$@CSH activated PMS for degradation of DC increased by 57% and 5.8 times compared to without catalyst. The efficiency of DC degradation by PMS alone was slightly improved due to the auxiliary activation of light and thermal effect. However, the ability of PBA/$MoS_2$@CSH activated PMS to degrade DC was increased by 60% under photoexcitation conditions, and the catalytic rate constant k increased by 7.2 and 1.6 times compared with Vis/

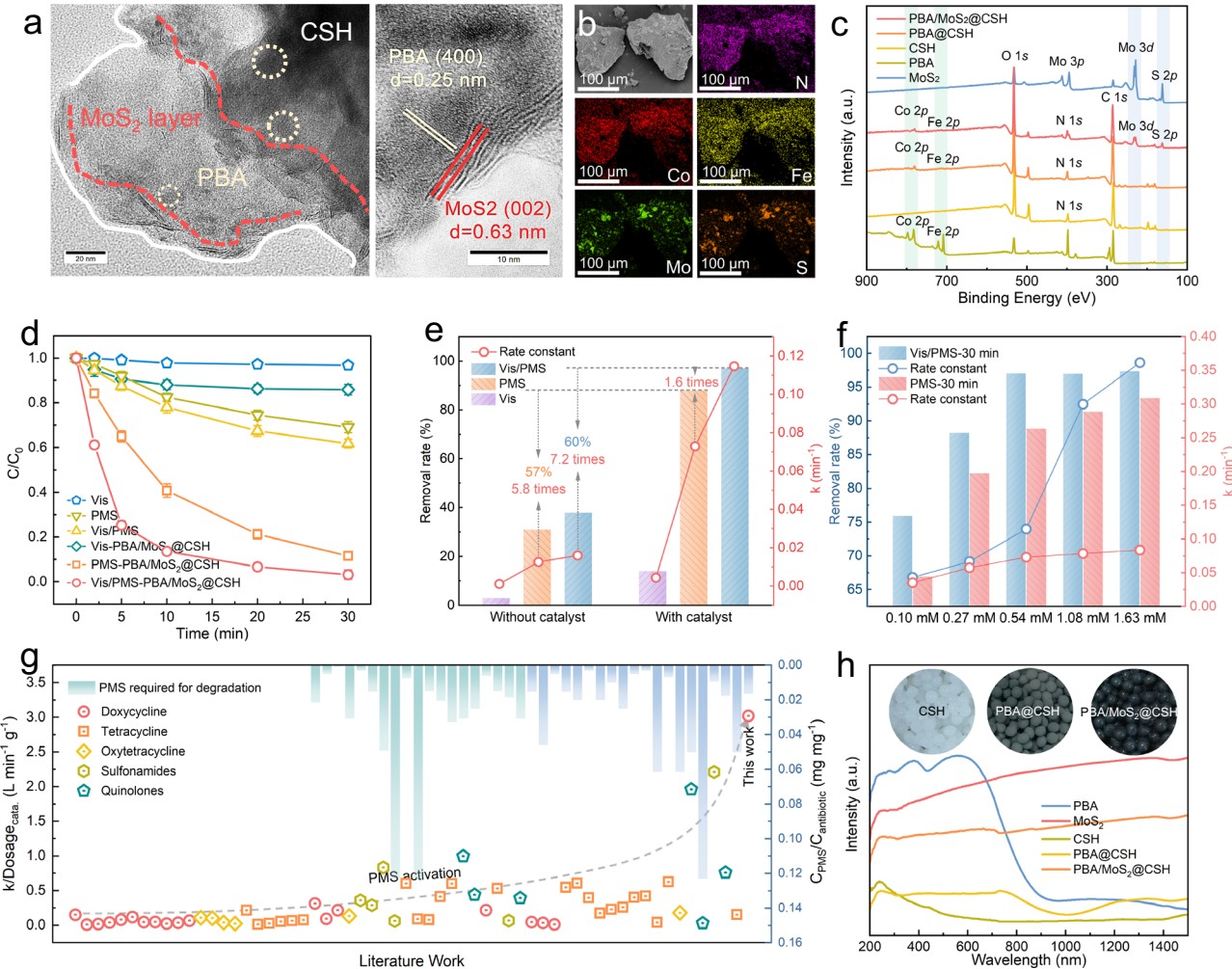

**Fig. 1 | Characterization and catalytic properties of PBA/MoS$_2$@CSH. a, b** TEM, HR−TEM images and element mapping of PBA/MoS$_2$@CSH. **c** Full XPS spectra of catalysts (Blue and green highlights indicate the positions of Mo 3$d$, S 2$p$ and Co 2$p$, Fe 2$p$, respectively). **d, e** Catalytic degradation curve and the relationship between photoexcitation (Vis: visible light) and PMS on degradation of DC. Error bars represent the standard deviation of the experiment in triplicate. **f** Difference of removal rate and apparent reaction rate constant k with or without photoexcitation at different PMS concentrations by PBA/MoS$_2$@CSH. **g** Comparisons of catalytic efficiency with those of reported catalysts. Literature comparison is provided in Supplementary Table 2. **h** UV−vis diffuse reflectance spectra and photo of catalysts.

PMS, and PMS − PBA/MoS$_2$@CSH. This may be related to the synergistic effect between photoexcited PBA/MoS$_2$ heterojunction and PMS activation. However, the reaction rate of catalytic activation with different PMS concentrations and DC removal ability were also significantly enhanced under photoexcitation compared with the catalyst alone (Fig. 1f and Supplementary Fig. 7). Compared with other 58 reported catalysts for antibiotic degradation, PBA/MoS$_2$@CSH had best catalyst-dose-rate constant k (L min$^{-1}$ g$^{-1}$) at low PMS/antibiotic concentration ratio (Fig. 1g and Supplementary Table 2). At the same time, the coupling system with the synergistic participation of light and PMS had a faster catalytic rate compared to the individual photocatalytic and PMS activation techniques. This result indicated the importance of the intervention of photoexcitation conditions in the activation of PMS and catalytic reaction by PBA/MoS$_2$@CSH. Therefore, the optical properties of the catalysts were explored through UV−vis diffuse reflectance spectra (Fig. 1h). The enhanced absorption of PBA@CSH optical region compared to CSH may be attributed to the absorption edge exhibited by PBA at 900 nm. In addition, PBA/MoS$_2$@CSH exhibited the comprehensive characteristics of PBA and MoS$_2$, showing stronger light absorption ability compared to PBA@CSH. This may be due to the wider spectrum response ability of MoS$_2$, the darker color, and the importance of heterostructures[11].

## Exploration of the driving force of interface electric field

The band structure of the catalyst was studied through density functional theory (DFT) to further explain the differences in activity (Supplementary Fig. 8). The calculated energy bands of MoS$_2$ (E$_g$ = 1.27 eV) and PBA (E$_g$ = 1.77 eV) were basically consistent with those in the literature[10,33] and those calculated by the Kubelka−Munk (K − M) equation (where the band gaps of MoS$_2$ and PBA were 1.25 and 1.60 eV, respectively) (Supplementary Fig. 9). The total density of states (DOS) calculated in Fig. 2a showed that the top of the VB of PBA/MoS$_2$ heterostructure was mainly composed of Co and Fe, and the shift towards the near Fermi level increases the VB maximum value compared with MoS$_2$ and PBA, while the bottom of the CB was mainly composed of Mo and S. Therefore, more electrons can be photoexcited to the CB due to easier electron transition, and the number of conducting carriers in PBA/MoS$_2$ increased, thus improving the photocatalytic activity[34,35]. The work function (Φ) was another important parameter in the study of electron transport in heterogeneous semiconductors, which can be estimated from the energy difference between the vacuum and Fermi levels according to the electrostatic potential[36]. As shown in Fig. 2b, the work functions of MoS$_2$ and PBA were 5.88 and 6.83 eV, respectively, and electrons tends to migrate from MoS$_2$ with higher Fermi levels to PBA with lower Fermi levels when they were in contact with each other to balance the Fermi energy of the two-component levels[37]. As shown

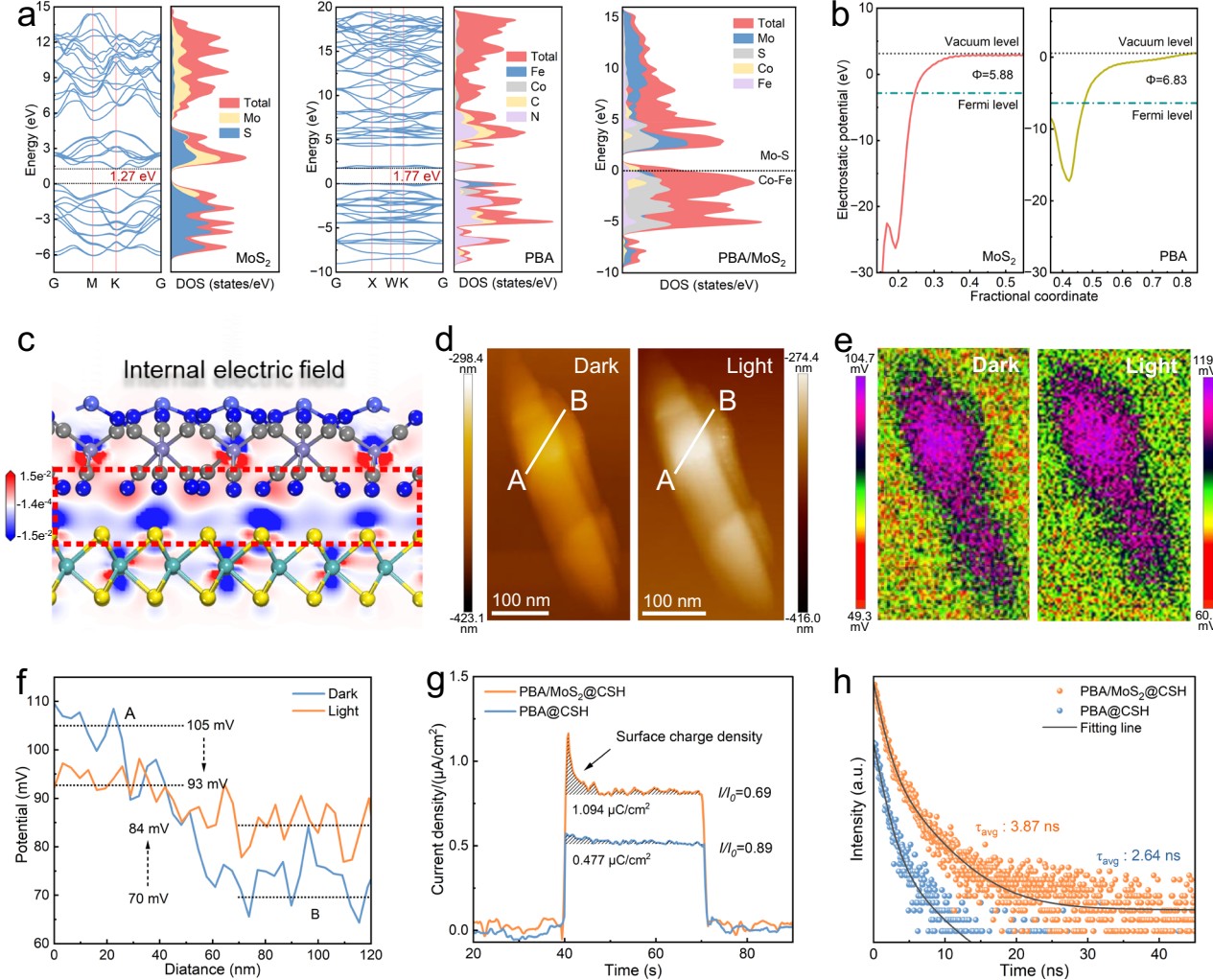

**Fig. 2 | Energy band structure and interface electric field properties. a** The energy band and PDOS of MoS$_2$, PBA, and the optimized configurations of PBA/MoS$_2$. **b** Electrostatic potentials of MoS$_2$ and PBA. **c** The charge difference distribution of PBA/MoS$_2$ interface, red and blue indicate charge accumulation and depletion. **d, e** Atomic force microscopy image and corresponding surface potential distribution of KPFM for PBA/MoS$_2$@CSH in darkness and under light irradiation. **f** The line-scanning surface potential from point A to B. **g, h** The surface charge density and TRPL spectroscopy of PBA@CSH and PBA/MoS$_2$@CSH, respectively.

in the differential charge distribution diagram in Fig. 2c, there was a strong interaction at the PBA/MoS$_2$ interface, resulting in the accumulation of more electrons to generate an interface electric field (IEF) to accelerate the separation of charge carriers. In order to further understand the charge transfer pathway of the IEF between MoS$_2$ and PBA, photoirradiated Kelvin probe force microscopy (KPFM) was applied (Fig. 2d and e). The surface potential difference between point A (MoS$_2$) and point B (PBA) was ~35 mV before light irradiation (Fig. 2f), indicating the formation of an IEF directed from point A to point B, which can serve as a driving force for photogenerated charge transfer. Upon visible light irradiation, the surface potential at point A significantly decreased, while the surface potential at point B increased. The change of interface surface potential revealed that the PBA in the heterojunction was an electron donor under illumination, as the electrons of B migrated to A, leading to an increased B potential[38,39].

The driving force of interfacial charge transfer and surface charge density were measured using the model developed by Kanata et al.[40], and the steady−state integral of transient photocurrent density measured over time. As shown in Fig. 2g, the surface charge density of PBA/MoS$_2$@CSH was significantly higher than that of PBA@CSH, while $i/i_0$ less than 1 indicated charge accumulation at the electrode[41]. According to Supplementary Fig. 10 and Supplementary Note 6, IEF of PBA/

MoS$_2$@CSH was 14.3 times that of PBA@CSH. Therefore, PBA/MoS$_2$@CSH also had better photoelectric responsiveness, smaller EIS arc radius and faster electron transfer rate (Supplementary Fig. 11). The separation and migration ability of photogenerated carriers were studied using steady−state photoluminescence (PL) and time-resolved photoluminescence (TRPL) spectra. Fluorescence was usually produced when the excited state of the light−captured material returned to the ground state. The PL emission peak intensity of PBA/MoS$_2$@CSH decreased significantly after the introduction of MoS$_2$ to form heterojunction, which demonstrated that IEF interaction between catalysts improved the photogenerated carrier separation efficiency (Supplementary Fig. 12). Meanwhile, the charge carrier lifetime had been extended from 2.64 ns of PBA@CSH to 3.87 ns of PBA/MoS$_2$@CSH (Fig. 2h and Supplementary Table 3). The PL and TRPL results indicated that the IEF played a crucial role in promoting the migration of photocarriers, slowing down the decay rate, and prolonging the lifetime. However, two important issues needed to be addressed: (i) how IEF promoted charge separation; (ii) How electrons and holes were transferred at the heterojunction interface by IEF. Therefore, we further explored real-time photogenerated charge dynamics using femtosecond transient absorption spectra (TAS). The photoexcitation of PBA/MoS$_2$@CSH produced a ground-state bleach

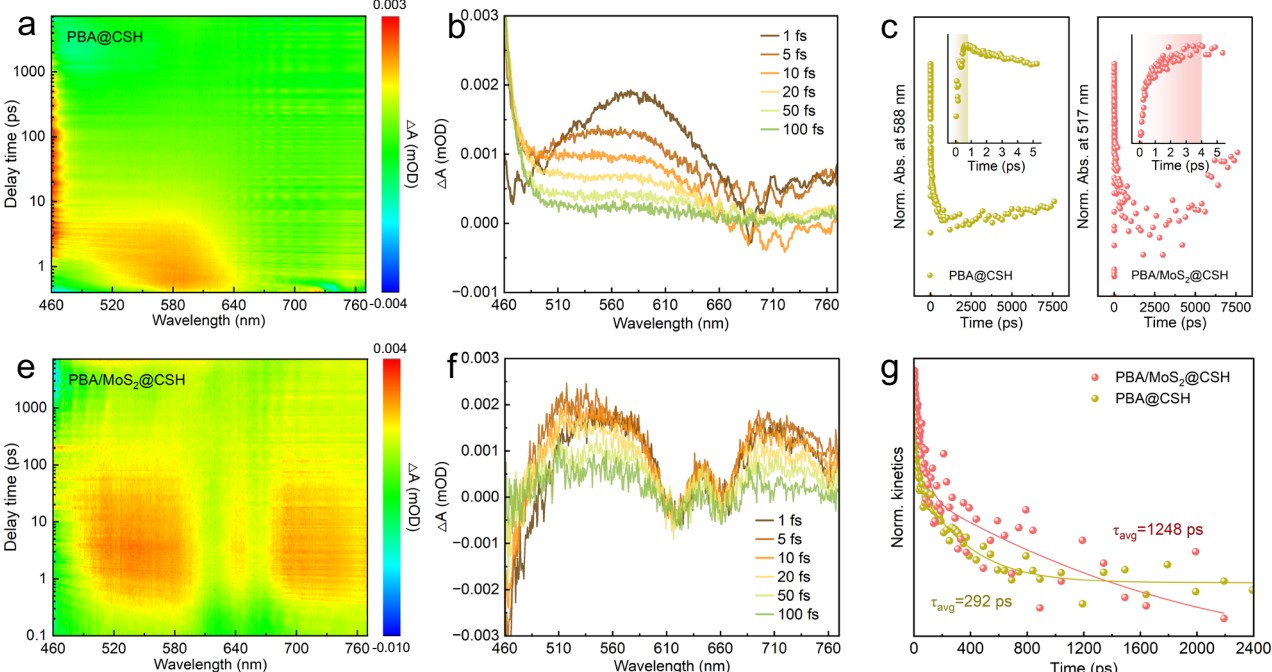

**Fig. 3 | Real-time photogenerated charge dynamics. a, b** Transient absorption spectra of PBA@CSH and **e, f** PBA/MoS₂@CSH. **c, g** Kinetic traces as a function of probe delay time and their corresponding normalized traces after 400 nm laser pulse irradiation (Highlighting indicates the time required to establish the ESA signal).

peak (GSB) at 460 nm and a positive excited-state absorption (ESA) peak at 517 nm, which can be allocated to the electronic transition of singlet state $S_1$ to $S_N$ (Fig. 3e and f)[42], and this signal also appeared in the PBA@CSH at 588 nm (Fig. 3a and b). In addition, two stimulated emission (SE) peaks at 618 and 660 nm can be observed in PBA/MoS₂@CSH, which was the signal generated by the rapid electron–hole recombination caused by the heterojunction interface through IEF, but the SE in PBA@CSH was not significant. We also found that the ESA peak in PBA/MoS₂@CSH was more significant and another peak appeared at 700 nm, which may be attributed to the component of MoS₂ in the heterojunction. Meanwhile, it can be observed that the establishment of ESA signal for PBA@CSH required 1 ps, but PBA/MoS₂@CSH required a longer time (Fig. 3c inset), which further confirmed the strong IEF, leading to rapid recombination of electron holes. It was worth noting that the kinetic trajectory curve of PBA/MoS₂@CSH slowly rose again, as the delay time was prolonged. This was because IEF drove the electrons of PBA from the separation site to the MoS₂ interface, and the free electrons captured in the impurity state were then slowly released. As observed, PBA/MoS₂@CSH still maintained a maximum electron survival rate of 60% after a charge decay of 7500 ps[43]. In addition, we also fitted the time distribution of TAS detected at 588 and 517 nm on PBA@CSH and PBA/MoS₂@CSH to estimate the decay kinetics of photo-generated carriers (Fig. 3g). The fitting parameters were listed in Supplementary Table 4, where $\tau_1$ corresponded to the process of electron capture, and long life $\tau_2$ was due to the interface electron transferred process. $\tau_3$ could be attributed to the recombination of holes and impurity electrons, while A referred to the proportion of photo-generated electrons involved[44]. Compared to the $\tau_2$ (20.15 ps), $A_2$ (0.155) and $\tau_3$ (310.49 ps), $A_3$ (0.236) of PBA@CSH, the $\tau_2$ (52.84 ps), $A_2$ (0.399) and $\tau_3$ (1322.48 ps), $A_3$ (0.310) of PBA/MoS₂@CSH were much larger. This result indicated that IEF opened a new channel for the transfer of photogenerated electrons from PBA to MoS₂. Therefore, the $\tau_{avg}$ carrier lifetime of PBA/MoS₂@CSH had been greatly extended to 1248.32 ps, providing sufficient evidence for the enhanced photocatalytic performance of IEF, which was consistent with PL, TRPL, and photoelectrochemical tests.

## S−scheme heterojunction carrier transfer mechanism

Identification of PMS activation active centers was critical to understanding the electron transfer process. Therefore, XPS was performed to analyze the valence state changes and catalytic mechanism of the photoexcited PMS system (Fig. 4a and b). The Mo⁴⁺ active site exposed by MoS₂ achieved high−speed electron transfer from the MoS₂ to PBA due to the work function. It accelerated the regeneration and cycling of Fe²⁺/Co²⁺⇌Co³⁺/Fe³⁺, greatly promoting the activation of PMS to form free radicals SO•₄⁻, •OH and ¹O₂ (Fig. 4c)[45]. In fact, the essence of Co³⁺ and Fe³⁺ reduction was to receive excess electrons. Therefore, the shifted of Co 2p and Fe 2p peaks to lower binding energy after PMS activation and the transition in valence state (Fig. 4a and Supplementary Fig. 13) were observed, and the presence of high valence Mo⁵⁺ and Mo⁶⁺ indicated the donation of electrons in the process (Fig. 4b)[46]. In situ irradiation XPS was also performed to observe the migration of photoexcited electrons. When light was applied to the PBA/MoS₂@CSH during measurement, compared to PMS in the dark, the peak of Co 2p and Fe 2p shifted positively, while the peak of Mo 3d shifted negatively with time. This result further proved the direction of photoelectron transfer from PBA to MoS₂. However, the change in the binding energy of XPS after photoexcitation/PMS activation may be attributed to the charge in the interface region generated by the coupling interaction between PBA and MoS₂ (strong electron interaction and chemical bonding)[47]. Therefore, the combination of band gap and VB−XPS directly determined the CB and VB positions of semiconductors to further explore the path and direction of carrier transport during photoexcitation (Fig. 4d and Supplementary Fig. 14). The CB and VB values of PBA were −0.83 and 0.77 eV, while those of MoS₂ were −1.58 and −0.33 eV, respectively. The difference of Fermi energy levels shown in the work functions of MoS₂ and PBA calculated by DFT determined that IEF must occur at the interface, which was also consistent with the results in the previous section. Therefore, when the electrons were continuously input into the PBA interface, the MoS₂ side was positively charged due to the consumption of electrons, whereas the PBA side was negatively charged due to the accumulation of electrons to balance the Fermi energy level. However, when PBA was

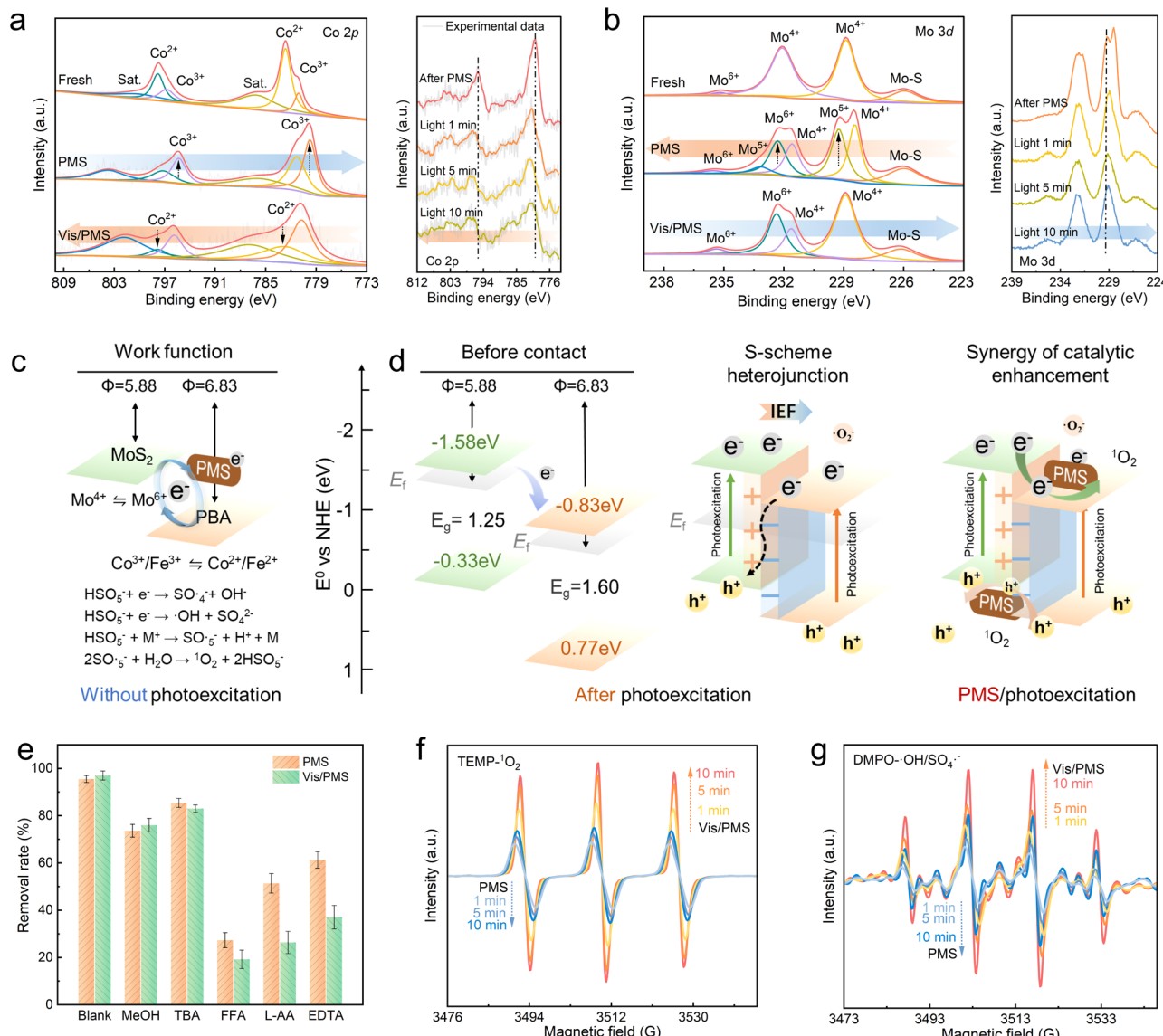

**Fig. 4 | Identification of S−scheme heterojunction/PMS activation active centers. a, b** The Co 2p and Mo 3d high−resolution XPS spectra of PBA/MoS₂@CSH before and after PMS, light or Vis/PMS tests, orange and blue arrows indicate the direction of peak shift, respectively. **c** Schematic diagram of PMS activation mechanism by PBA/MoS₂@CSH transition metal and **d** PBA/MoS₂@CSH S−scheme heterojunction with interface electric field. **e** Comparison of the removal rate under different quenchers. Error bars indicate standard deviation of three measurements of quenching experiment. **f, g** EPR signals for DMPO− • OH/SO₄ • ⁻ and TMPO−¹O₂.

excited by light to produce photogenerated electrons from VB, the IEF and Coulombic attraction of holes generated by MoS₂, as well as band bending were all beneficial for PBA electrons to be driven to the VB of MoS₂ to suppress their recombination with holes in PBA space. This path was used to coordinate photogenerated carriers that migrate to the surface to participate in PMS activation and DC degradation. Therefore, Mo 3d will also obtain certain electrons in the S−scheme heterostructure, leading to the increase of electron density after the Vis/PMS reaction. On the contrary, the electron density of Co 2p and Fe 2p decreases, showing a shift in the direction of higher binding energy[48]. Therefore, S−scheme heterojunction can vividly describe the photogenerated carrier transfer path and confer higher redox capacity on PBA/MoS₂@CSH[21,49]. The results of radical quenching experiments shown that DC degradation involved a variety of pathways, including ¹O₂ and •OH, SO • ₄⁻ and •O₂⁻ assisted degradation (Fig. 4e). However, the contribution of •O₂⁻ and holes had a certain increase after the participation of photoexcitation. The generation of ¹O₂ depended on charge transfer or energy transfer. •O₂⁻ could be oxidized by holes or disproportionated to ¹O₂[50]. In PMS alone system, PBA/MoS₂@CSH was

used as an electron donor or acceptor to undergo redox conversion with PMS to produce ROS. However, in the Vis/PMS system, PBA/MoS₂@CSH was regarded as an S−scheme heterojunction, the photoexcited electrons undergo redox conversion with PMS through an IEF to generate ROS. Simultaneously, photogenerated holes oxidized PMS to generate SO•₅⁻ and ¹O₂ or directly affect the degradation of DC. Therefore, multiple synergistic pathways greatly enhanced the ability of ROS generation and significantly increased the degradation rate of DC. The EPR experiment further verified the synergistic activation of S−scheme heterojunction and PMS. As shown in Fig. 4f and g, the ability of PBA/MoS₂@CSH activated PMS to produce ¹O₂ and •OH/SO • ₄⁻ under photoexcitation was significantly improved at different times compared with the PMS alone. This provided strong evidence for the synergistic driving of PMS activation by photoexcitation S−scheme heterojunction with IEF to enhance the ROS production.

**Evaluation of reaction pathways and catalytic performance**
The catalytic reaction involved the following steps: PMS was captured by the catalyst to form surface complexes, and S−scheme

heterojunction was excited by light to generate photogenerated carriers, carriers were separated and transported to the surface to participate in catalytic reactions, and PMS was activated to degrade DC through an S−scheme heterojunction mediated electron transfer mechanism. It can be seen from Fig. 5a and Supplementary Fig. 15 that pH had little influence on the degradation of DC by photoexcitation PBA/MoS$_2$@CSH activated PMS. The inhibition of catalytic rate at the initial pH of 3.16 was because PMS was more stable and difficult to be activated under acidic conditions[51]. The improvement in removal efficiency as the initial pH increases to 10.97 may be attributed to alkali-activated PMS. Interestingly, in the PBA/MoS$_2$@CSH synergistic PMS activation reaction, the solution pH was maintained between 2.7 and 3.7, which coincided with the pH$_{IEP}$ range of PBA/MoS$_2$@CSH positive charge (Fig. 5b). As was well known, PMS existed in the form of anions over a wide pH range, and HSO$_5^-$ was the main type of PMS (pK$_{a1}$ = 0.4, pK$_{a2}$ = 9.3)[52]. Therefore, PMS can be captured by protonated chitosan due to electrostatic attraction, which results in an accelerated reaction rate. It was worth noting that the restricted structure also affected the activation pathway of PMS[53]. The constraint effect of the chitosan spontaneous entanglement network constructed a 3D pathway for PBA/MoS$_2$, which allowed electrons to transfer through the IEF in the heterojunction to accelerate the transport of photogenerated electrons and holes[54]. In-situ FTIR measurements showed that the stretching vibration intensity of the S−O bond decreased significantly at 1204 and 1108 cm$^{-1}$ for HSO$_5^-$ and SO$_4^{2-}$ (Fig. 5c). This should be the result of rapid PMS decomposition. However, no shift in the absorption band of the S−O bond of HSO$_5^-$ had been observed, indicating that the O−O bond was directly broken rather than the generation of metastable M$^{n+}$−(HO) OSO$_3^-$ during PMS

activation[55,56]. Therefore, the catalytic reaction path between PMS and PBA/MoS$_2$@CSH was predicted by DFT (Supplementary Fig. 16). In Step 1, PMS was first captured by protonated chitosan 3D hydrogel (the $E_{ads}$ of PMS adsorbed on protonation chitosan was −0.29 eV, indicating that the reaction can be spontaneous) (Fig. 5d). However, PMS was transferred to the PBA/MoS$_2$ interface through Step 2 for synchronous activation due to its higher $E_{ads}$ = −5.15 eV (Fig. 5e). At the same time, the superposition vibration enhancement of N−H and Fe−CN−Co peaks in the FTIR spectrum after reaction also proved the participation of active sites in the structure (Supplementary Fig. 17). More importantly, the charge density difference illustrated the apparent accumulation and depletion of charge in the PMS molecule and PBA/MoS$_2$ interface, which confirmed the occurrence of the rapid electron transfer and PMS activation (Step 3). In addition, the O−O bond ($l_{O-O}$) of the adsorbed PMS was extended from 1.390 to 1.467 Å, indicating that the PMS molecules were activated and ready to be cleaved to produce ROS (Fig. 5f and g)[57]. The activation of PMS and the use of electron-dense regions generated by photoexcited heterojunction to induce the DC degradation through free/non−free radicals involved electron migration. Therefore, the electron transfer mechanism between PBA/MoS$_2$@CSH with PMS and photoexcitation was analyzed through chronoamperometry and chronopotentiometry (Supplementary Fig. 18). The addition of PMS greatly increased the current output and the potential also increased sharply, indicating that the exchange of electrons between PMS and PBA/MoS$_2$@CSH induced a high redox potential. However, the addition of photoexcitation led to a further increase in the intensity, which showed the enhancement mechanism of the S−scheme heterojunction interface electric field, and the further change of current and potential during the addition of

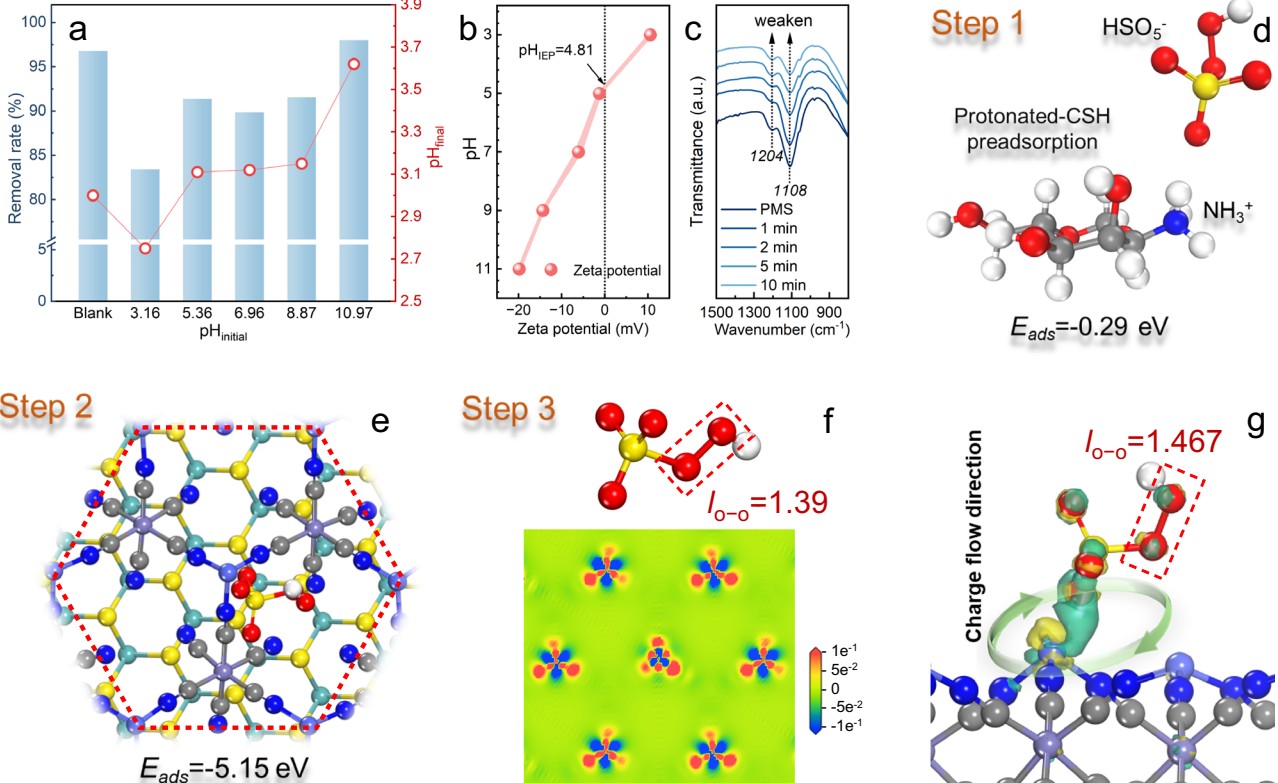

**Fig. 5 | Reaction mechanism and steps of PMS activation by PBA/MoS$_2$@CSH. a** The removal rate of DC and pH$_{final}$ and **b** zeta potential of PBA/MoS$_2$@CSH at different pH. Error bars represent the standard deviation of the test in triplicate. **c** In situ FTIR variation of PMS with reaction time. **d−g** The catalytic reaction route Step 1: PMS was pre−captured by protonation chitosan to form surface complex (**d**);

Step 2: PMS was transferred to PBA/MoS$_2$ S−scheme heterojunction due to higher $E_{ads}$ (**e**); Step 3: electron transfer and PMS activation were demonstrated by charge density difference (**f** and **g**). Yellow and green indicate charge depletion and accumulation, respectively.

DC demonstrated the progress of catalytic reaction[58]. All in all, the mechanism of photoexcited PBA/MoS$_2$@CSH synergistic PMS activation was as follows. At the microscopic level, the IEF of the S−scheme heterojunction redistributed the energy barrier and accelerated electron transport, and then changed the electronic properties and mechanism of transition metal-activated PMS. Therefore, the synergistic reaction of multiple activation paths was conducive to the accumulation of excess charge around the PMS, resulting in an enhanced catalytic reaction rate. From a macro perspective, the spontaneous entanglement network space of chitosan hydrogel enabled faster enrichment of pollutants and oxidants. The amount of activator could be reduced due to the attraction of protonated chitosan to PMS, which could make the PMS act on the active site faster and reduce the movement of short−lived free radicals in space to accelerate the redox reaction. Therefore, photoexcited PBA/ MoS$_2$@CSH synergistic PMS activation fundamentally changed the traditional transition metal redox reaction pathway, and enhanced the synergistic activation of PMS by S−scheme heterojunction with IEF, which contributed to the efficient catalytic degradation of DC.

DC degradation was evaluated by three simulated light sources (sunlight, visible light, LED) (Fig. 6a) to explore future application of PBA/MoS$_2$@CSH. The difference in degradation performance of DC may be influenced by the spectral range and power size of the light source, but it still had a significant degradation effect even through 5 W white LED. It was well known that temperature rise was conducive to molecular diffusion and PMS decomposition (endothermic reaction)[59]. Thus, continuous control of low temperature can affect degradation efficiency (Supplementary Fig. 19a), which also corresponded to the reason for the increase in DC degradation rate of PBA/MoS$_2$@CSH under wide spectrum and high−power conditions. Therefore, the thermal effect, the spectral range of the light source, and the initial pH of the aqueous solution all had a certain impact on the activation rate of PMS (Fig. 6b). As can be seen from Supplementary Fig. 19b, PBA/MoS$_2$@CSH still had good performance in the treatment of DC (40 mg L$^{-1}$) with higher concentration (it was worth noting that five hydrogel spheres used in each experiment were equivalent to only 1.15 mg transition metal catalyst in idealized state in Supplementary Fig. 2c, possibly less). Therefore, PBA/MoS$_2$@CSH had a higher removal rate and a fairly fast

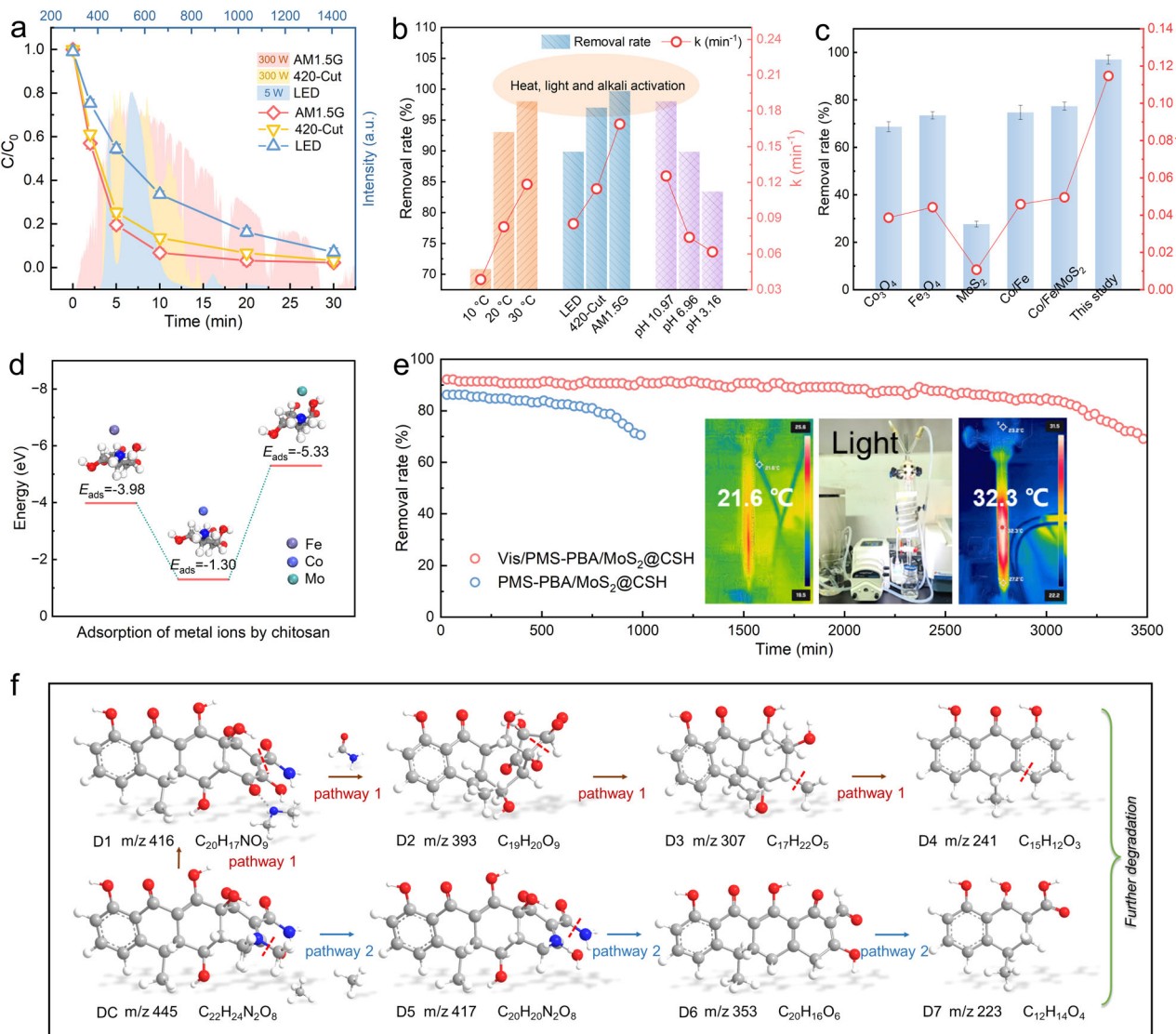

**Fig. 6 | Evaluation of catalytic performance and DC degradation path. a** The degradation of DC under different illuminant. **b** The efficiency relationship between temperature, illuminant and pH on DC catalytic degradation. **c** Comparison of catalytic efficiency using commercial catalysts with this study. Error bars in a and c represent the standard deviation of the experiment in triplicate. **d** $E_{ads}$ of interaction between Co, Fe and Mo and chitosan. **e** Performance of PBA/MoS$_2$@CSH in continuous flow reactions (inset was photothermal camera photo). **f** The possible pathways for DC degradation by PBA/MoS$_2$@CSH under Vis/PMS.

catalytic efficiency compared with the same amount of commercial catalysts, while having DC mineralization capacity, with a total organic carbon (TOC) removal rate of 74% in 30 min (Fig. 6c and Supplementary Fig. 20). Meanwhile, chitosan hydrogel entangled network space had a good affinity for transition metals (Fig. 6d)[60]. Therefore, the metal ions leached from the solution after degradation were much lower than those directly using CoFe/MoS$_2$ powder and other reported catalysts (Supplementary Fig. 21 and Supplementary Table 5 and 6), as well as the Chinese National Standard (GB25467 – 2010)[61]. This showed that PBA/MoS$_2$@CSH was very stable in the reaction, and the spherical hydrogel beads could be easily recovered after the reaction without residue, which would not cause secondary pollution to the environment. PBA/MoS$_2$@CSH showed different degradation rates for the three classes of eight antibiotics, which may be the result of selective oxidation of $^1O_2$ produced in the process. However, all antibiotics can be continuously degraded, possibly due to the simultaneous presence of $^1O_2$, •OH, $SO•_4^-$, •$O_2^-$ and holes in the reaction, which can oxidize pollutants nonselectively (Supplementary Fig. 22). PBA/MoS$_2$@CSH exhibited the fastest DC degradation efficiency in deionized water due to the absence of active material scavenging or light shielding (Supplementary Fig. 23 and Supplementary Note 7). The degradation of DC in environmental water was slightly inhibited, especially in wastewater inflow (municipal water) due to the removal of organic and inorganic substances. These results indicated that the photoactivation PBA/MoS$_2$@CSH synergistic PMS system used to improve DC problems was most suitable for advanced oxidation in the later stage of water treatment. Continuous flow experiments were carried out to further study the stability of PBA/MoS$_2$@CSH in consideration of cost and practicability under LED light (Supplementary Fig. 24). It can be observed from Fig. 6e that Vis/PMS – PBA/MoS$_2$@CSH system can degrade DC for up to 3000 min, significantly longer than that of using PBA/MoS$_2$@CSH as a transition metal activator (the removal efficiency begins to decline after about 600 min). This better removal performance and stability were mainly attributed to the synergistic PMS catalytic degradation of DC through various activation pathways mediated by S–scheme heterojunction with IEF. Meanwhile, we found that the inevitable thermal effect of its black hydrogel beads would also play an auxiliary role, as shown in Fig. 6b. The characterization analysis before and after the stability experiment proved that PBA/MoS$_2$@CSH had good stability (Supplementary Fig. 25). The intermediates and possible pathways and mechanisms for DC degradation were further investigated by HPLC – MS and DFT (Fig. 6f and Supplementary Fig. 26, and Supplementary Table 7). The Fukui electrophilic index ($f^-$) and the highest occupied molecular orbital (HOMO) of DC in the methyl region near N$_{25}$ indicated that it was easy to lose electrons and be attacked by electrophilic reagents (holes and $^1O_2$) (Supplementary Fig. 27 and Supplementary Table 8)[21]. Therefore, DC may generate intermediates D1 and D5 through bond breaking or demethylation. A high radical index ($f^0$) indicated that the atom was easily attacked by $SO•_4^-$ and ·OH. Thus, the reaction further opened the ring and gradually broke the bond to form D2 to D6. The lowest unoccupied molecular orbital (LUMO) and nucleophilic index ($f^+$) in the hydroxyl functional region of O$_{20}$ indicated the possibility of •$O_2^-$ attacked[62]. So it was possible to form a series of intermediate products such as D7 through functional group cleavage, and then further oxidized the final intermediate to be mineralized into H$_2$O and CO$_2$. The developmental toxicity index of DC evaluated using Toxicity Estimation Software (T.E.S.T.) was 0.87, which could cause interference with larval development. However, the developmental toxicity values of most intermediates decreased during degradation (Supplementary Fig. 28). This result showed that PBA/MoS$_2$@CSH could not only efficiently degrade DC but also reduce its ecotoxicity.

## Discussion

In summary, we constructed PBA/MoS$_2$@CSH and proposed the mechanism of synergistic PMS activation through photoexcitation of S –scheme heterojunction. The electric field at the PBA/MoS$_2$ interface redistributed the energy barrier and provided the driving force for charge transport. The integration of photocarriers into the PMS synergistic activation system changed the electronic properties and structure of the traditional transition metal–activated PMS and enhanced the catalytic reaction rate through multiple activation paths. Simultaneously, chitosan hydrogel 3D polymer space enabled faster enrichment of PMS to reach the active sites faster, which accelerated the redox reaction and led to the degradation of DC via free radical/non–free radical. The experimental and DFT results analyzed the possible degradation pathways of DC, and proved that photoexcitation PBA/MoS$_2$@CSH synergistic activated PMS system had good stability for advanced oxidation in the later stage of water treatment. This study extends the application of traditional PMS activation technology to degrade emerging organic pollutants and provides new insights for the design of MOF–based S–scheme heterojunction photocatalytic synergistic activation of PMS functional materials.

## Methods
### Materials
All the chemicals in this study were commercially available for the catalysts preparation and catalytic performance evaluation and described in Supplementary Note 1.

### Synthesis of CoFe PBA
CoFe PBA was synthesized by the wet chemical precipitation method. In a typical experiment, CoCl$_2$·6H$_2$O (237.9 mg) and C$_6$H$_5$Na$_3$O$_7$ (658.5 mg) were dissolved in 20 ml of deionized water. Meanwhile, 658.5 mg of K$_3$[Fe(CN)$_6$] was also evenly dissolved in 20 mL of deionized water. Then uniformly mix the above solution at 25 °C for 12 h. The precipitate was collected, centrifuged and washed several times with deionized water and ethanol, and then dried overnight at 60 °C.

### Synthesis of hierarchical MoS$_2$
The hierarchical MoS$_2$ was synthesized by one–step solvothermal reactions. First, MoO$_3$ (0.44 g) and KSCN (1.2 g) were dissolved in 60 mL of deionized water, stirring and ultrasonic to dissolve fully. It is then kept in a Teflon–lined stainless–steel autoclave at 200 °C for 24 h. The obtained samples were washed and dried separately with deionized water and ethanol.

### Establishing S–scheme heterojunction chitosan hydrogel interface
First, PBA and MoS$_2$ were grind repeatedly in a mortar with a small amount of ethanol in a certain ratio, and then dried for standby. Chitosan hydrogel was prepared by a simple pH phase inversion method. Specifically, 1 g of chitosan was dissolved in 2% acetic acid (v/v, 40 mL). Then, PBA (0.2 g) /MoS$_2$ (0, 0.1, 0.2, 0.3 g) was added to the solution under mechanical agitation to obtain a uniform suspended colloid. It was added to (1.25 M NaOH/0.1 M CH$_3$COONa) solution by drops through a syringe and gelated for 3 h. The product was then washed in deionized water until neutral and dried at 35 °C. The collected hydrogels were called PBA@CSH, PBA/MoS$_2$@CSH–1, PBA/MoS$_2$@CSH–2 and PBA/MoS$_2$@CSH–3. The chitosan hydrogel without any catalyst was called CSH. The purpose of adding different amounts of MoS$_2$ was to find the best proportion, so that the final sample had the best performance.

### Analytical methods
The characterization methods employed in this study were described in Supplementary Note 2. The mechanism of photocatalytic activation of S–scheme heterojunction/PMS on DC degradation was investigated by photocatalytic and activated PMS experiments. Experimental

procedures and DFT calculations were detailed in Supplementary Note 3 and Supplementary Note 4 in the Support information.

## Data availability

The data that support the findings of this study are included in the published article (and its Supplementary Information) or available from the corresponding author upon request.

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

## Acknowledgements

We gratefully acknowledge the National Key Research and Development Program of China (No. 2022YFC3005802), the National Natural Science Foundation of China (No. 52070052), the TOUYAN Project of Heilongjiang Province (AUEA5640201520-01) and State Key Laboratory of Urban Water Resource and Environment (Harbin Institute of Technology) (No. 2022TS16). Additional thanks go to the High–Performance Computing Center of HIT and the Project of Young Scientist Studio of HIT. The authors would like to thank Jiayu Zhang and Xin Du from Xiyunyice Lab (www.sci-aide.com) for the in-situ-test analysis.

## Author contributions

A.W., D.M.L., and W.W. conceived and designed the experiments. A.W. and X.L. synthesized the materials and M.D. assisted the characterization. A.W. and J.N. performed the catalytic activity experiments. M.D., D.Q.L., and X.L. contributed to data analysis. J.M. and J.W. supervised the project. A.W. analyzed the results and wrote the paper. D.M.L., J.W. and W.W provided constructive suggestions for the manuscript revision. A.W., M.D. and Y. P. significantly revised the manuscript. All authors commented on the manuscript.

## Competing interests

The authors declare no competing interests.
