## [Peer review file · Nature Communications]

Enhanced and synergistic catalytic activation by photoexcitation driven S-scheme heterojunction hydrogel interface electric fieldREVIEWER COMMENTS

Reviewer #1 (Remarks to the Author):

This manuscript reported an interesting and important result, which realized the transformation of reaction mechanism in the photoexcitation driven PMS activation system for the advanced oxidation process through the synthesis of S-scheme heterojunction PBA/MoS₂@chitosan hydrogel. This process promoted the oxidation-reduction of PMS under the directional driving of the interface electric field within the heterojunction, strengthening the ability of ROS generation and significantly increasing the rate of antibiotic degradation. Many studies have reported the use of photocatalysts to promote Fenton like reactions, but this work proposes new explanations for this critical scientific issue. Interestingly, this hydrogel can not only be used as a conventional transition metal catalyst to activate PMS, but also realize the synergistic enhancement of heterogeneous catalysis oxidation system reaction under the photoexcitation, providing a new strategy for coupling the S-scheme heterojunction photocatalysis and PMS activation process. The manuscript is well written and figures are pretty and well designed, and the new mechanisms and phenomena have been well validated through complete experiments and logical representations. Therefore, it is recommended to publish in Nature Communications after minor revision.

1. English tenses may need to be standardized somewhere, such as: "heterojunction" or "heterojunctions" need to be unified throughout the entire manuscript. Part 1.5 " It is well known that temperature rise " "is" should be "was"...
2. The author provided a detailed description of the carrier transfer pathway in S-scheme heterojunctions, but in PMS based oxidation systems, I am curious about the advantages of S-scheme heterojunctions compared to other types I, II and the Z-scheme? What is the difference in this electron transfer method?
3. A near-IR absorption of 1500 nm was measured in the UV-vis diffuse reflectance spectra. Black samples, as well as the narrow bandgaps, can contribute to the near-IR adsorption. What role might this extra light absorption serve? Furthermore, Figure 1f was not mentioned in the manuscript.
4. The author said that this research has excellent catalytic performance when the amount of transition metal catalyst used is very small, and it is difficult to quantify through the figure. It is suggested to solve this problem by comparing with other heterogeneous catalysis oxidation systems or photocatalytic performance, so that the conclusion of its superior performance can be better supported.
5. Please clarify the novelty of this work compared with the recent published papers.

Reviewer #2 (Remarks to the Author):

This manuscript reported the synthesized S-scheme heterojunction PBA/MoS₂@chitosan hydrogel to achieve the photoexcitation synergistic PMS activation. The structure characterization and analysis, discussed by the authors for the PBA/MoS₂@chitosan hydrogel catalyst, are very common. Besides, the photoexcited carriers of constructed heterojunction undergo redox conversion with PMS through an excellent S-scheme transfer pathway driven by the directional interface electric field. However, the authors proposed that the characterization and analysis, for the formed built-in electric field in the photocatalyst, are very general in photocatalytic composites, which is not better than the reported references and cannot give some valuable inspirations for rational fabrication and better understanding the photocatalytic mechanism. Overall, the characterizations are not matching with the conclusions. Therefore, this work does not have enough suitable for publication in Nat. Commun. at present level. To enhance this work, I have few minor issues and also a more general question that would need to be addressed.

Detailed comments:

1. The quality of English need to be improved. The abbreviation (such as ROC), subscript and

superscripts should be re-checked.

2. The conclusion of internal electric field is not convincing. By observing the change in interface potential from Kelvin probe force microscopy (Adv. Mater. 2021, 33, 2100317; Angew. Chem. Int. Ed. 2022, 61, e202113044) upon photoirradiation rather than just electrochemical behaviors, it is better to confirm the directional interface electric field in PBA/MoS₂@chitosan hydrogel catalyst.

3. The Mo⁴⁺ active site exposed by MoS₂ achieved high-speed electron transfer from the MoS₂ to PBA due to the work function. It accelerated the regeneration and cycling of Fe²⁺/Co²⁺ \rightleftharpoons Co³⁺/Fe³⁺, greatly promoting the activation of PMS to form free radicals. To support the S-scheme heterojunction carrier transfer mechanism, operando time-resolved spectroscopy should be considered such as the time-resolved analysis of XAS, TAS or XPS.

4. Reaction mechanism and steps of PMS activation by PBA/MoS₂@CSH is given, but no experimental data confirm the possible intermediates. During DC photocatalytic degradation, the in-situ diffuse reflectance infrared Fourier transform spectra measurements should be performed for detecting the reaction intermediates.

5. Although the metal ions leached from the solution after degradation was lower than those directly using CoFe/MoS₂ powder, PBA/MoS₂@CSH still have a high Co, Fe and Mo contents in the degradation solution. From its practical application, it exists the secondary pollution to the environment.

6. Does the ratio of CoFe, MoS₂ and CSH in the heterojunction affect photocatalytic activities?

7. When discussing the built-in electric field in the photocatalyst, it is necessary to calculate the electron band structures (CB, VB).

Reviewer #3 (Remarks to the Author):

The authors demonstrated the S-scheme heterojunction PBA/MoS₂@chitosan hydrogel, which realized the photoexcitation synergistic PMS activation system by interface electric field. It should be the first job to use S-scheme heterojunction photocatalyst to enhance PMS activation. The novel approach remarkably improved the catalytic performance, which is believed to provide good inspiration for the coupling of photocatalysis and advanced oxidation. Besides, the experiment data was logical and plentiful and stated in a systemic way, and visualization of figures was high quality. This work should be interesting to the readership of Nature Communications. However, there are some issues to be addressed, this manuscript can be accepted after minor revisions.

1. How is the charge transfer and hole mobility of S-scheme heterojunction? What is the difference with Type-I, Type-II, and Z-scheme?

2. Some figures have not mentioned in the manuscript, like Figure 1f, 3f, 4b and 5f (Please clarify, unless I overlooked).

3. Many typos should be addressed such as "Eg" should be revised "Eg(subscript)", including Figure S8 and S13

4. The authors should compare the photocatalysis or PMS activation performance in this manuscript with those of reported catalytic materials.

5. Some statements in results and discussion should be supported with relevant references, such as: Page 7 "The C≡N vibration peak ... Because there were different coordination modes in the PBA structure" Page 20 "The Fukui electrophilic index.... indicated that it was easy to lose electrons and be attacked by electrophilic reagents"...

6. Some minor English linguistic errors and formatting issues are present. Double-check the grammar for the whole manuscript.

Response to Reviews

Thank you for the reviewers' comments concerning our manuscript. Those comments are valuable and helpful for improving our paper, as well as the important guiding significance to our researches. We have studied comments carefully and have made correction which we hope meet with approval.

Revised portion are marked in **Highlight** in the paper.

Reviewer #1:

This manuscript reported an interesting and important result, which realized the transformation of reaction mechanism in the photoexcitation driven PMS activation system for the advanced oxidation process through the synthesis of S-scheme heterojunction PBA/MoS₂ @chitosan hydrogel. This process promoted the oxidation-reduction of PMS under the directional driving of the interface electric field within the heterojunction, strengthening the ability of ROS generation and significantly increasing the rate of antibiotic degradation. Many studies have reported the use of photocatalysts to promote Fenton like reactions, but this work proposes new explanations for this critical scientific issue. Interestingly, this hydrogel can not only be used as a conventional transition metal catalyst to activate PMS, but also realize the synergistic enhancement of heterogeneous catalysis oxidation system reaction under the photoexcitation, providing a new strategy for coupling the S-scheme heterojunction photocatalysis and PMS activation process. The manuscript is well written and figures are pretty and well designed, and the new mechanisms and phenomena have been well validated through complete experiments and logical representations. Therefore, it is recommended to publish in Nature Communications after minor revision.

Q1. English tenses may need to be standardized somewhere, such as: “heterojunction” or “heterojunctions” need to be unified throughout the entire manuscript. Part 1.5 " It is well known that temperature rise " "is" should be "was"...

Response: *We thank the reviewer's comment and made the following modifications:*

In Page 2 Line 26 (Abstract): “There is reason to believe that the synergistic activation of PMS by S–scheme heterojunctions regulated...” has been modified as “There is reason to believe that the synergistic activation of PMS by S–scheme heterojunction regulated”

In Page 4 Line 66 (Introduction): “Among the candidates for constructing heterojunctions, molybdenum disulfide (MoS₂), a layered transition metal chalcogenide, exhibits great potential in PMS activation and catalysis” has been modified as “Among the candidates for constructing heterojunction, molybdenum disulfide (MoS₂), a layered transition metal chalcogenide, exhibits great potential in PMS activation and catalysis”

In Page 10 Line 162 (**Results**): “The total density of states (DOS) calculated in Figure 2a shown that the top of the **valence band (VB)** of PBA/MoS₂” **has been modified as** “The total density of states (DOS) calculated in **Fig. 2a** shown that the top of the **VB** of PBA/MoS₂”

In Page 10 Line 165 (**Results**): “while the bottom of the **conduction band (CB)** was mainly composed of Mo and S.” **has been modified as** “while the bottom of the **CB** was mainly composed of Mo and S.”

In Page 12 Line 202 (**Results**): “Fluorescence was usually produced when the excited state of the light-captured material **returns** to the ground state...” **has been modified as** “Fluorescence was usually produced when the excited state of the light-captured material **returned** to the ground state...”

In Page 12 Line 204 (**Results**): “...after the introduction of MoS₂ to form heterojunction, which **mean** that IEF interaction between catalysts...” **has been modified as** “...after the introduction of MoS₂ to form heterojunction, which **demonstrated** that IEF interaction between catalysts...”

In Page 16 Line 274 (**Results**): “Therefore, S-scheme **heterojunctions** can vividly describe the photogenerated carrier transfer path and confer higher redox capacity on PBA/MoS₂@CSH” **has been modified as** “Therefore, S-scheme **heterojunction** can vividly describe the photogenerated carrier transfer path and confer higher redox capacity on PBA/MoS₂@CSH”

In Page 16 Line 274 (**Results**): “This **provides** strong evidence for the synergistic driving of PMS activation by photoexcitation S-scheme heterojunction with IEF to enhance the ROS production.” **has been modified as** “This **provided** strong evidence for the synergistic driving of PMS activation by photoexcitation S-scheme heterojunction with IEF to enhance the ROS production.”

In Page 18 Line 300 (**Results**): “...and S-scheme **heterojunctions** was excited by light to generate photogenerated carriers...” **has been modified as** “and S-scheme **heterojunction** was excited by light to generate photogenerated carriers,”

In Page 18 Line 310 (**Results**): “the solution pH was maintained between 2.7 and 3.7, which **coincides** with the pH_{IEP} range of PBA/MoS₂@CSH positive charge” **has been modified as** “the solution pH was maintained between 2.7 and 3.7, which **coincided** with the pH_{IEP} range of PBA/MoS₂@CSH positive charge”

In Page 19 Line 331 (**Results**): “The activation of PMS and the use of electron dense regions generated by photoexcited **heterojunctions** to induce the DC degradation through free/non-free radicals involved electron migration” **has been modified as** “The activation of PMS and the use of electron dense regions generated by photoexcited **heterojunction** to induce the DC degradation through free/non-free radicals involved electron migration”

In Page 21 Line 364 (**Part 3.5**): “**It is well known that** temperature rise was conducive to molecular

diffusion and PMS decomposition (endothermic reaction)” **has been modified as** “**It was well known that** temperature rise was conducive to molecular diffusion and PMS decomposition (endothermic reaction)”

Q2. The author provided a detailed description of the carrier transfer pathway in S-scheme heterojunctions, but in PMS based oxidation systems, I am curious about the advantages of S-scheme heterojunctions compared to other types I, II and the Z-scheme? What is the difference in this electron transfer method?

Response: We thank the reviewer’s comment. We have made modifications in the manuscript and supporting information. First, please allow me to explain the differences between various heterojunction.

Typically, there are two types of conventional heterojunction photocatalysts, those with a straddling gap (type-I), and those with a staggered gap (type-II) [1]. For the type-I heterojunction photocatalyst (see Fig. 1a), the conduction band (CB) and the valence band (VB) of semiconductor A are respectively higher and lower than the corresponding bands of semiconductor B. Therefore, under light irradiation, the electrons and holes will accumulate at the CB and the VB levels of semiconductor B, respectively. Since both electrons and holes accumulate on the same semiconductor, the electron–hole pairs cannot be effectively separated for the type-I heterojunction photocatalyst. Moreover, a redox reaction takes place on the semiconductor with the lower redox potential, thereby significantly reducing the redox ability of the heterojunction photocatalyst. For the type-II heterojunction photocatalyst (see Figure 1b), the CB and the VB levels of semiconductor A are higher than the corresponding levels of the semiconductor B. Thus, the photogenerated electrons will transfer to semiconductor B, while the photogenerated holes will migrate to semiconductor A under light irradiation, resulting in a spatial separation of electron–hole pairs. Similar to the type-I heterojunction, the redox ability of the type-II heterojunction photocatalyst will be also reduced because the reduction reaction and the oxidation reaction take place on semiconductor B with lower reduction potential and on semiconductor A with lower oxidation potential, respectively.

Figure 1 Schematic illustration of the two different types of separation of electron–hole pairs in the case of

conventional light-responsive heterojunction photocatalysts: a) type-I, and b) type-II [1].

To mimic natural photosynthesis in plants and to enhance redox ability of photocatalysts, traditional Z-scheme photocatalysts were proposed by Bard in 1979, which can improve charge-separation efficiency and also reserve strong redox ability. This system is constructed of two semiconductors with suitable intermediate couples. These two semiconductors have staggered band structure configurations. In an ideal process, photogenerated holes in the VB of PC I react with electron donors (D), yielding the corresponding electron acceptors (A). Whereas the photogenerated electrons in the CB of PC II react with A, engendering D. Then, the reserved photogenerated electrons in the CB of PC I and holes in the VB of PC II participate in the reduction and oxidation reactions, respectively (Figure 2). Such a charge-transfer mode can render this system with strong redox ability, together with spatially separated redox reaction sites [2]. However, this Z-scheme heterojunction still requires another medium to act as A and D.

Figure 2. Schematic illustration of charge transfer in traditional Z-scheme heterojunction photocatalysts [2].

Based on the understanding of traditional heterojunction, a new concept of S-scheme heterojunction is proposed. This concept was first proposed by Professor Jiaguo Yu in 2019 [2][3]. Its full name is step-scheme heterojunction. This heterojunction is mainly composed of a reduced semiconductor photocatalyst (RP) with a smaller work function and a higher Fermi level and an oxidized semiconductor photocatalyst (OP) with a larger work function and a lower Fermi level (Figure 3). It is constructed in this type manner, which can effectively realize the separation of electron-hole pairs with strong redox ability. An S-scheme heterojunction is composed of RP and OP with staggered band structures, which is similar to type-II heterojunction but with a completely different charge-transfer route. In a typical type-II heterojunction, photogenerated electrons and holes are accumulated on the CB of OP and VB of RP, respectively, resulting in weak redox ability. In an S-scheme heterojunction, the powerful photogenerated electrons and holes are reserved in the CB of RP and VB of OP, respectively, while the pointless photogenerated charge carriers are

recombined, introducing a strong redox potential. In addition, it can be seen that the charge-transfer route in S-scheme mode resembles “step” in macroscopic (from low CB to high CB) and letter of N in microscopic. This is where it gets its name from.

Figure 3. (a) The work functions of $g\text{-C}_3\text{N}_4$ and WO_3 before contact. (b) The internal electric field and band edge bending at the interface of $\text{WO}_3/g\text{-C}_3\text{N}_4$ after contact. (c) The S-scheme charge transfer mechanism between WO_3 and $g\text{-C}_3\text{N}_4$ under light irradiation [3].

[1] J. Low, J. Yu, M. Jaroniec, S. Wageh, A.A. Al-Ghamdi, *Heterojunction Photocatalysts*, *Advanced Materials*, 29 (2017) 1601694.

[2] Q. Xu, L. Zhang, B. Cheng, J. Fan, J. Yu, *S-Scheme Heterojunction Photocatalyst*, *Chem*, 6 (2020) 1543-1559.

[3] J. Fu, Q. Xu, J. Low, C. Jiang, J. Yu, *Ultrathin 2D/2D $\text{WO}_3/g\text{-C}_3\text{N}_4$ step-scheme H_2 -production photocatalyst*, *Applied Catalysis B: Environmental*, 243 (2019) 556-565.

We have made the following modifications in the manuscript:

In Page 4 Line 59 (**Introduction**): “The recently developed S-scheme heterojunction has an efficient charge transfer ability (Fig. S1 and Text S6)²¹. Meanwhile, electrons and holes generated by photoexcitation heterojunction accumulate at the more negative conduction band (CB) and positive valence band (VB) positions, respectively. Therefore, the S-scheme heterojunction enhances charge separation and redox capabilities.”

Added Figure S1 and Text S6 to the support information:

Figure S1. Schematic illustration of charge-carrier transfer pathways of conventional type-II (a), S-scheme (b) and Z-scheme heterojunction (c).

Text S6. The advantages of S-scheme heterojunction

The S-scheme heterojunction is composed of reduced semiconductor photocatalyst (RP) and oxidized semiconductor photocatalyst (OP) with a staggered band structure, which is similar to the conventional type-II heterojunction but has completely different charge transfer routes. For type-II heterojunction, the photo generated electrons of the RP with higher CB are transferred from the interface to the OP with lower CB, while the photo generated holes of the OP with lower VB migrate to the RP with higher VB, resulting in a decrease in the redox ability of the system (Figure S1a). On the contrary, the S-scheme heterojunction photogenerated carriers accumulate on the RP at higher CB positions and the OP at lower VB positions, thereby enhancing the charge separation and redox ability of the charge carriers (Figure S1b). Therefore, compared with the traditional type I and type II heterostructures, the S-scheme heterojunction improves the redox ability, and compared with the Z-type heterostructure, it does not need the medium of electron transfer at the interface (Figure S1c).

Q3. A near-IR absorption of 1500 nm was measured in the UV-vis diffuse reflectance spectra. Black samples, as well as the narrow bandgaps, can contribute to the near-IR adsorption. What role might this extra light absorption serve? Furthermore, Figure 1f was not mentioned in the manuscript.

Response: It can be observed that a single PBA exhibits absorption edges in the range of 800nm to 900nm, but it is not stable at 800nm. Therefore, near-IR measurements were conducted in the 800-1500nm range to compare the light absorption ability of the catalysts. Indeed, darker samples and narrower bandgaps have better light absorption and lower reflectivity to light. This study evaluated the degradation of DC in hydrogel beads under visible light, LED and AM1.5G under experimental conditions. The absorption of near-IR light should to some extent improve the degradation of DC by catalysts in sunlight or actual environments.

Figure 1f. UV–vis diffuse reflectance spectra and photo of catalysts.

In Page 9 Line 151 (**Results**): “Therefore, the optical properties of the catalysts were explored through UV–vis diffuse reflectance spectra (DRS).” **has been modified as** “Therefore, the optical properties of the catalysts were explored through UV–vis diffuse reflectance spectra (Fig. 1h).”

Q4. The author said that this research has excellent catalytic performance when the amount of transition metal catalyst used is very small, and it is difficult to quantify through the figure. It is suggested to solve this problem by comparing with other heterogeneous catalysis oxidation systems or photocatalytic performance, so that the conclusion of its superior performance can be better supported.

Response: We thank the reviewer’s comment and made the following modifications:

In Page 9 Line 145 (**Results**): “Compared with other 58 reported catalysts for antibiotic degradation, PBA/MoS₂@CSH had excellent catalyst-dose-rate constant k ($L \text{ min}^{-1} \text{ g}^{-1}$) at low PMS/antibiotic concentration ratio (Fig. 1g and Table S2). At the same time, the coupling system with the synergistic participation of light and PMS had a faster catalytic rate compared to the individual photocatalytic and PMS activation techniques.”

Fig.1g and Table S2 has been added to the manuscript and supporting information.

Fig. 1 g Comparisons of catalytic efficiency with those of reported catalysts.

Table S2. Comparison with other photocatalysts in literature.

Photocatalyst	Pollutant	Light source	PMS	Dosage	Removal Time	k (min ⁻¹)	Reference
PBA/MoS ₂ @CSH	Doxycycline (20 mg L ⁻¹)	300W XL ($\lambda \geq 420$ nm)	0.333 g L ⁻¹	0.038 g L ⁻¹	97.02% 30 min	0.1147	This study
LaFeO ₃ /SBA-15	Doxycycline (40 mg L ⁻¹)	300 W XL ($\lambda \geq 420$ nm)	0.615 g L ⁻¹	0.500 g L ⁻¹	/	0.023	2
BiFeO ₃ /SBA-15	Doxycycline (40 mg L ⁻¹)	300 W XL ($\lambda \geq 420$ nm)	1.844 g L ⁻¹	0.500 g L ⁻¹	/	0.0175	2
BiO _{1-x} Cl	Doxycycline (50 mg L ⁻¹)	5 W LED ($\lambda \geq 400$ nm)	0.250 g L ⁻¹	0.500 g L ⁻¹	79.4 105	0.0062	3
g-C ₃ N ₄ /Na-BiVO ₄	Tetracycline (20 mg L ⁻¹)	300 W XL ($\lambda \geq 420$ nm)	0.307 g L ⁻¹	0.200 g L ⁻¹	98.2% 40 min	0.109	4
Bi ₂ MoO ₆ /CuWO ₄	Tetracycline (10 mg L ⁻¹)	300 W XL ($\lambda \geq 420$ nm)	0.200 g L ⁻¹	0.200 g L ⁻¹	84.6% 20 min	0.121	5
10%Co ₃ O ₄ /g-C ₃ N ₄	Tetracycline (20 mg L ⁻¹)	300 W XL ($\lambda \geq 420$ nm)	0.061 g L ⁻¹	0.200 g L ⁻¹	98% 60 min	0.079	6
BC/CN-15	Tetracycline (10 mg L ⁻¹)	300 W XL ($\lambda \geq 420$ nm)	0.200 g L ⁻¹	0.200 g L ⁻¹	90% 60 min	0.035	7
Cu - R	Tetracycline (30 mg L ⁻¹)	300 W XL ($\lambda \geq 420$ nm)	0.300 g L ⁻¹	0.200 g L ⁻¹	96% 60 min	0.046	8
MnCo ₂ O ₄	Tetracycline (30 mg L ⁻¹)	300 W XL ($\lambda \geq 420$ nm)	0.750 g L ⁻¹	0.200 g L ⁻¹	98% 60 min	0.052	9

Photocatalyst	Pollutant	Light source	PMS	Dosage	Removal Time	k (min ⁻¹)	Reference
3NiO/g-C ₃ N ₄	Tetracycline (20 mg L ⁻¹)	300 W XL ($\lambda \geq 420$ nm)	0.100 g L ⁻¹	0.200 g L ⁻¹	/ 60 min	0.080	10
MCN	Tetracycline (10 mg L ⁻¹)	300W XL ($\lambda \geq 420$ nm)	0.500 g L ⁻¹	0.400 g L ⁻¹	89.7 30	0.061	11
MoS ₂ /Ag/g-C ₃ N ₄	Tetracycline (20 mg L ⁻¹)	300 W XL ($\lambda \geq 420$ nm)	0.061 g L ⁻¹	0.200 g L ⁻¹	91.2% 30 min	0.084	12
MoO ₃ /Bi ₂ O ₃ /g-C ₃ N ₄	Tetracycline (40 mg L ⁻¹)	Solar light	2.459 g L ⁻¹	0.600 g L ⁻¹	98% 120min	0.0248	13
CuHNPs-7.5	Tetracycline (40 mg L ⁻¹)	100W LED ($\lambda \geq 420$ nm)	0.277 g L ⁻¹	0.200 g L ⁻¹	97.8% 30 min	0.125	14
FeMo ₃ O ₃ /C ₃ N ₄ -EP	Oxytetracycline (50 mg L ⁻¹)	300 W XL ($\lambda \geq 420$ nm)	3.074 g L ⁻¹	1.000 g L ⁻¹	98.1	0.181	15
SMM-3	Levofloxacin (10 mg·L ⁻¹)	300 W XL ($\lambda \geq 420$ nm)	0.500 g L ⁻¹	0.100 g L ⁻¹	95.1% 20 min	0.196	16
Ag/AgCl@ZIF-8/g-C ₃ N ₄	Levofloxacin (10 mg·L ⁻¹)	150 W XL ($\lambda \geq 420$ nm)	1.230 g L ⁻¹	1.000 g L ⁻¹	87.3% 60 min	0.03054	17
Fe ₃ O ₄ @CeO ₂ @BiOI	Sulfamethoxazole (13 mg·L ⁻¹)	UVA-LED	0.123 g L ⁻¹	0.100 g L ⁻¹	97% 15 min	0.221	18
γ -Fe ₂ O ₃ -MnO ₂	Ciprofloxacin (17 mg·L ⁻¹)	300 W XL ($\lambda \geq 420$ nm)	0.300 g L ⁻¹	0.150 g L ⁻¹	98.3% 30 min	0.114	19
CoCr ₂ O ₄ / α -Fe ₂ O ₃ / β -La ₂ S ₃	Doxycycline (10 mg L ⁻¹)	1000W HL ($\lambda \geq 420$ nm)	/	0.050 g L ⁻¹	92.83% 345 min	0.0076	20

Photocatalyst	Pollutant	Light source	PMS	Dosage	Removal Time	k (min ⁻¹)	Reference
Co/Mn-MOF-74@g-C ₃ N ₄	Doxycycline (40 mg L ⁻¹)	300 W XL ($\lambda \geq 420$ nm)	/	0.500 g L ⁻¹	/	0.00459	21
g-C ₃ N ₄ / α -Bi ₂ (MoO ₄) ₃	Doxycycline (10 mg L ⁻¹)	500 W XL ($\lambda \geq 420$ nm)	/	1.000 g L ⁻¹	93.19% 140 min	0.0183	22
In ₂ O ₃ /Bi ₄ O ₇	Doxycycline (20 mg L ⁻¹)	300W XL ($\lambda \geq 420$ nm)	/	0.500 g L ⁻¹	92.1% 120 min	0.0197	23
AN@CN	Doxycycline (50 mg L ⁻¹)	300W XL ($\lambda \geq 420$ nm)	/	0.500 g L ⁻¹	98.67% 60 min	0.04052	24
BiM/ZnC@PANI	Doxycycline (10 mg L ⁻¹)	300W XL ($\lambda \geq 420$ nm)	/	0.100 g L ⁻¹	90% 150 min	0.0119	25
ILDAC/MIL-68(In)-NH ₂	Doxycycline (10 mg L ⁻¹)	500 W XL ($\lambda \geq 420$ nm)	/	0.200 g L ⁻¹	92% 180 min	0.00918	26
ZnO	Doxycycline (10 mg L ⁻¹)	30 W UV-C lamp	/	0.250 g L ⁻¹	~88% 780 min	0.012	27
Bi7O9I3/g-C3N4	Doxycycline (20 mg L ⁻¹)	300W XL ($\lambda \geq 420$ nm)	/	0.500 g L ⁻¹	80% 120 min	0.0125	28
BiOBr/FeWO ₄	Doxycycline (20 mg L ⁻¹)	300 W XL ($\lambda \geq 420$ nm)	/	1.000 g L ⁻¹	90.4% 60 %	0.0375	29
Nd-BiO _{2-x}	Doxycycline (10 mg L ⁻¹)	300 W XL ($\lambda \geq 420$ nm)	/	0.200 g L ⁻¹	86.14% 120 min	0.01344	30
Co ₃ O ₄ TiO ₂ /GO	Oxytetracycline (10 mg L ⁻¹)	300 W XL ($\lambda \geq 400$ nm)	/	0.250 g L ⁻¹	91% 90 min	0.0272	31

Photocatalyst	Pollutant	Light source	PMS	Dosage	Removal Time	k (min ⁻¹)	Reference
Ag/p-Ag ₂ S/n-BiVO ₄	Oxytetracycline (20 mg L ⁻¹)	500 W XL ($\lambda \geq 420$ nm)	/	0.400 g L ⁻¹	99.8% 150 min	0.0411	32
AgI/BiVO ₄	Oxytetracycline (20 mg L ⁻¹)	300 W XL ($\lambda \geq 420$ nm)	/	0.600 g L ⁻¹	80% 60 min	0.0239	33
SrTiO ₃ /BiOI	Oxytetracycline (20 mg L ⁻¹)	300 W XL ($\lambda \geq 420$ nm)	/	1.000 g L ⁻¹	85.34% 90 min	0.0252	34
Ag/N-GQDs/g-C ₃ N ₄	Tetracycline (10 mg L ⁻¹)	300 W XL ($\lambda \geq 365$ nm)	/	0.200 g L ⁻¹	92.8% 60 min	0.0428	35
CQDs/g-C ₃ N ₄	Tetracycline (10 mg L ⁻¹)	250 W XL ($\lambda \geq 420$ nm)	/	0.500 g L ⁻¹	78.6% 240 min	0.00642	36
h-BN/g-C ₃ N ₄	Tetracycline (10 mg L ⁻¹)	300 W XL ($\lambda \geq 420$ nm)	/	1.000 g L ⁻¹	79.7% 60 min	0.02775	37
h-BN/Bi ₂ MoO ₆	Tetracycline (20 mg L ⁻¹)	300W XL ($\lambda \geq 420$ nm)	/	0.500 g L ⁻¹	99.19% 140 min	0.0273	38
NGQDs-BiOI/MnNb ₂ O ₆	Tetracycline (10 mg L ⁻¹)	250 W XL ($\lambda \geq 420$ nm)	/	0.500 g L ⁻¹	87.2% 60 min	0.0331	39
TiO ₂ @V ₂ O ₅ -PPy	Tetracycline (50 mg L ⁻¹)	300W XL ($\lambda \geq 420$ nm)	/	0.600 g L ⁻¹	96% 60 min	0.04498	40
Mg-Fe LDH@bioachar	Doxycycline (35 mg L ⁻¹)	/	0.750 g L ⁻¹	0.750 g L ⁻¹	88.76% 120 min	0.23571	41
MnO/CoO/WO ₃	Doxycycline (20 mg L ⁻¹)	/	0.100 g L ⁻¹	0.500 g L ⁻¹	80.04% 120 min	0.0471	42

Photocatalyst	Pollutant	Light source	PMS	Dosage	Removal Time	k (min ⁻¹)	Reference
CuO/Fe ₂ O ₃	Doxycycline (50 mg L ⁻¹)		0.050 g L ⁻¹	0.200 g L ⁻¹	92.6% 120 min	0.04342	43
FeVO ₄ nanorods	Oxytetracycline (20 mg L ⁻¹)		0.615 g L ⁻¹	0.800 g L ⁻¹	100% 30 min	0.107	44
Co-Fe/NC ^{0.7} @GCS	Sulfamethoxazole (20 mg·L ⁻¹)		0.0615 g L ⁻¹	0.200 g L ⁻¹	90.2% 60 min	0.072	45
NSC-3	Sulfamethoxazole (20 mg·L ⁻¹)		0.307 g L ⁻¹	0.200 g L ⁻¹	98.62% 90 min	0.058	46
MF	Sulfamethoxazole (10 mg·L ⁻¹)		0.492 g L ⁻¹	0.060 g L ⁻¹	100% 80min	0.050	47
BC700Fe20	Sulfamethoxazole (10 mg·L ⁻¹)		1.230 g L ⁻¹	0.500 g L ⁻¹	82.2% 120min	0.031	48
Co ₃ O ₄ /CPANI	Tetracycline (20 mg L ⁻¹)		0.150 g L ⁻¹	0.150 g L ⁻¹	92.11% 40 min	0.09033	49
Fe ₃ O ₄ @PANI-p	Tetracycline (20 mg L ⁻¹)		2.459 g L ⁻¹	0.400 g L ⁻¹	89.8% 90 min	0.0353	50
PFSC-900	Tetracycline (20 mg L ⁻¹)		0.300 g L ⁻¹	0.400 g L ⁻¹	90.91 120	0.0317	51
Goethite/biochar	Tetracycline (30 mg L ⁻¹)		0.615 g L ⁻¹	0.050 g L ⁻¹	72.99 60	0.02062	52
Co-Ni LDO	Tetracycline (30 mg L ⁻¹)		0.984 g L ⁻¹	0.100 g L ⁻¹	100 60	~0.06	53

Photocatalyst	Pollutant	Light source	PMS	Dosage	Removal Time	k (min ⁻¹)	Reference
CoFe ₂ O ₄	Levofloxacin (5 mg L ⁻¹)		0.154 g L ⁻¹	0.100 g L ⁻¹	94.63 30	0.0997	54
CA-LDH	Ciprofloxacin (20 mg L ⁻¹)		0.500 g L ⁻¹	0.200 g L ⁻¹	98 60	0.088	55
Co@N-BC	Doxycycline (50 mg L ⁻¹)		0.307 g L ⁻¹	0.400 g L ⁻¹	92.72 30	0.0873	56
Cu-In ₂ O ₃ /O _v	Tetracycline (20 mg L ⁻¹)		0.300 g L ⁻¹	0.500 g L ⁻¹	100 20	0.2648	57
EGCG@Fe ₃ O ₄	Sulfadiazine (10 mg L ⁻¹)		0.184 g L ⁻¹	0.800 g L ⁻¹	97.9 60	0.0541	58
SBC ₈₀₀	Norfloxacin (10 mg L ⁻¹)		0.307 g L ⁻¹	0.200 g L ⁻¹	100 40	0.0785	59

- Zhao Q, Long M, Li H, Wen Q, Li D. Synthesis of MFeO₃/SBA-15 (M = La or Bi) for peroxymonosulfate activation towards enhanced photocatalytic activity. *New Journal of Chemistry* 46, 1144–1157 (2022).
- Liu M, et al. Confine activation peroxymonosulfate by surface oxygen vacancies of BiO_{1-x}Cl to boost its utilization rate. *Separation and Purification Technology* 307, 122711 (2023).
- Kang J, et al. The enhanced peroxymonosulfate-assisted photocatalytic degradation of tetracycline under visible light by g-C₃N₄/Na-BiVO₄ heterojunction catalyst and its mechanism. *Journal of Environmental Chemical Engineering* 9, 105524 (2021).
- Chen R, Dou X, Xia J, Chen Y, Shi H. Boosting peroxymonosulfate activation over Bi₂MoO₆/CuWO₄ to rapidly degrade tetracycline: Intermediates and mechanism. *Separation and Purification Technology* 296, 121345 (2022).
- Jin C, et al. Two dimensional Co₃O₄/g-C₃N₄ Z-scheme heterojunction: Mechanism insight into enhanced peroxymonosulfate-mediated visible light photocatalytic performance. *Chemical Engineering Journal* 398, 125569 (2020).
- Tang R, et al. π-π stacking derived from graphene-like biochar/g-C₃N₄ with tunable band structure for photocatalytic antibiotics degradation via peroxymonosulfate activation. *Journal of Hazardous Materials* 423, 126944 (2022).
- Yi L, Li Y, Zhu L, Gao C, Wu X. CuO decorated natural rectorite as highly efficient catalyst for photoinduced peroxymonosulfate activation towards tetracycline degradation. *Journal of Cleaner Production* 317, 128441 (2021).
- Wang J, Jiang Y, Gao C, Li Y, Wu X. Synergistic effect of bimetal in three-dimensional hierarchical MnCo₂O₄ for high efficiency of photoinduced Fenton-like reaction. *Surfaces and Interfaces* 27, 101482 (2021).
- Li S, et al. NiO/g-C₃N₄ 2D/2D heterojunction catalyst as efficient peroxymonosulfate activators toward tetracycline degradation: Characterization, performance and mechanism. *Journal of Alloys and Compounds* 880, 160547 (2021).

11. Shi H, He Y, Li Y, He T, Luo P. Efficient degradation of tetracycline in real water systems by metal-free g-C₃N₄ microspheres through visible-light catalysis and PMS activation synergy. *Separation and Purification Technology* 280, 119864 (2022).
12. Jin C, Kang J, Li Z, Wang M, Wu Z, Xie Y. Enhanced visible light photocatalytic degradation of tetracycline by MoS₂/Ag/g-C₃N₄ Z-scheme composites with peroxymonosulfate. *Applied Surface Science* 514, 146076 (2020).
13. Alnaggar G, Hezam A, Dmash QA, Ananda S. Sunlight-driven activation of peroxymonosulfate by microwave synthesized ternary MoO₃/Bi₂O₃/g-C₃N₄ heterostructures for boosting tetracycline hydrochloride degradation. *Chemosphere* 272, 129807 (2021).
14. Guo T, Jiang L, Huang H, Li Y, Wu X, Zhang G. Enhanced degradation of tetracycline in water over Cu-doped hematite nanoplates by peroxymonosulfate activation under visible light irradiation. *Journal of Hazardous Materials* 416, 125838 (2021).
15. Liu Y, et al. Enhanced activation of peroxymonosulfate by a floating FeMo₃O_x/C₃N₄ photocatalyst under visible-light assistance for oxytetracycline degradation: Performance, mechanisms and comparison with H₂O₂ activation. *Environmental Pollution* 316, 120668 (2023).
16. He Y, et al. Acceleration of levofloxacin degradation by combination of multiple free radicals via MoS₂ anchored in manganese ferrite doped perovskite activated PMS under visible light. *Chemical Engineering Journal* 431, 133933 (2022).
17. Zhou J, Liu W, Cai W. The synergistic effect of Ag/AgCl@ZIF-8 modified g-C₃N₄ composite and peroxymonosulfate for the enhanced visible-light photocatalytic degradation of levofloxacin. *Science of The Total Environment* 696, 133962 (2019).
18. Kohantorabi M, Moussavi G, Oulego P, Giannakis S. Radical-based degradation of sulfamethoxazole via UVA/PMS-assisted photocatalysis, driven by magnetically separable Fe₃O₄@CeO₂/BiOI nanospheres. *Separation and Purification Technology* 267, 118665 (2021).
19. Zhao J, Wang Y, Li N, Wang S, Yu J, Li X. Efficient degradation of ciprofloxacin by magnetic γ-Fe₂O₃-MnO₂ with oxygen vacancy in visible-light/peroxymonosulfate system. *Chemosphere* 276, 130257 (2021).
20. Sivarajani PR, et al. Fabrication of ternary nano-heterojunction via hierarchical deposition of α-Fe₂O₃ and β-La₂S₃ on cubic CoCr₂O₄ for enhanced photodegradation of doxycycline. *Journal of Industrial and Engineering Chemistry* 118, 407-417 (2023).
21. Wen Q, et al. Synergistic effect of photocatalysis and peroxymonosulfate activated by Co/Mn-MOF-74@g-C₃N₄ Z-scheme photocatalyst for removal of tetracycline hydrochloride. *Separation and Purification Technology* 313, 123518 (2023).
22. Vasanthakumar V, et al. α-Bi₂(MoO₄)₃ nanorods decorated with two-dimensional g-C₃N₄ nanosheets for efficient degradation of doxycycline under visible light illumination. *Process Safety and Environmental Protection* 163, 1-13 (2022).
23. Pan Z, Qian L, Shen J, Huang J, Guo Y, Zhang Z. Construction and application of Z-scheme heterojunction In₂O₃/Bi₄O₇ with effective removal of antibiotic under visible light. *Chemical Engineering Journal* 426, 130385 (2021).
24. Feng C, et al. Core-shell Ag₂CrO₄/N-GQDs@g-C₃N₄ composites with anti-photocorrosion performance for enhanced full-spectrum-light photocatalytic activities. *Applied Catalysis B: Environmental* 239, 525-536 (2018).
25. Wang A, et al. MOF derived ZnO clusters on ultrathin Bi₂MoO₆ yolk@shell reactor: Establishing carrier transfer channel via PANI tandem S-scheme heterojunction. *Applied Catalysis B: Environmental* 328, 122492 (2023).
26. Li D, et al. In-situ fabrication of ionic liquids/MIL-68(In)-NH₂ photocatalyst for improving visible-light photocatalytic degradation of doxycycline hydrochloride. *Chemosphere* 292, 133461 (2022).
27. Pourmoslemi S, Mohammadi A, Kobarfard F, Amini M. Photocatalytic removal of doxycycline from aqueous solution using ZnO nano-particles: a comparison between UV-C and visible light. *Water science and technology : a journal of the International Association on Water Pollution Research* 74, 1658-1670 (2016).
28. Zhang Z, Pan Z, Guo Y, Wong PK, Zhou X, Bai R. In-situ growth of all-solid Z-scheme heterojunction photocatalyst of Bi₇O₉I₃/g-C₃N₄ and high efficient degradation of antibiotic under visible light. *Applied Catalysis B: Environmental* 261, (2020).
29. Gao J, Gao Y, Sui Z, Dong Z, Wang S, Zou D. Hydrothermal synthesis of BiOBr/FeWO₄ composite photocatalysts and their photocatalytic degradation of doxycycline. *J Alloy Compd* 732, 43-51 (2018).
30. Wang Q, et al. Unsaturated Nd-Bi dual-metal sites enable efficient NIR light-driven O₂ activation for water purification. *Applied Catalysis B: Environmental* 319, 121924 (2022).
31. Jo W-K, Kumar S, Isaacs MA, Lee AF, Karthikeyan S. Cobalt promoted TiO₂/GO for the photocatalytic degradation of oxytetracycline and Congo Red. *Appl Catal B-Environ* 201, 159-

32. Wei Z, et al. A novel 3D plasmonic p-n heterojunction photocatalyst: Ag nanoparticles on flower-like p-Ag₂S/n-BiVO₄ and its excellent photocatalytic reduction and oxidation activities. *Appl Catal B-Environ* 229, 171–180 (2018).
33. Guan DL, Niu CG, Wen XJ, Guo H, Deng CH, Zeng GM. Enhanced *Escherichia coli* inactivation and oxytetracycline hydrochloride degradation by a Z-scheme silver iodide decorated bismuth vanadate nanocomposite under visible light irradiation. *J Colloid Interf Sci* 512, 272–281 (2018).
34. Wen XJ, Niu CG, Zhang L, Liang C, Zeng GM. An in depth mechanism insight of the degradation of multiple refractory pollutants via a novel SrTiO₃/BiOI heterojunction photocatalysts. *J Catal* 356, 283–299 (2017).
35. Peng Y, et al. Construction of Plasmonic Ag and Nitrogen-Doped Graphene Quantum Dots Decorated Ultrathin Graphitic Carbon Nitride Nanosheet Composites with Enhanced Photocatalytic Activity: Full-Spectrum Response Ability and Mechanism Insight. *ACS Appl Mater Inter* 9, 42816–42828 (2017).
36. Hong Y, et al. Facile fabrication of stable metal-free CQDs/g-C₃N₄ heterojunctions with efficiently enhanced visible-light photocatalytic activity. *Sep Purif Technol* 171, 229–237 (2016).
37. Jiang L, et al. Metal-free efficient photocatalyst for stable visible-light photocatalytic degradation of refractory pollutant. *Appl Catal B-Environ* 221, 715–725 (2018).
38. Du Z, et al. Ultrathin h-BN/Bi₂MoO₆ heterojunction with synergetic effect for visible-light photocatalytic tetracycline degradation. *J Colloid Interface Sci* 589, 545–555 (2021).
39. Yan M, et al. Fabrication of nitrogen doped graphene quantum dots-BiOI/MnNb₂O₆ p-n junction photocatalysts with enhanced visible light efficiency in photocatalytic degradation of antibiotics. *Appl Catal B-Environ* 202, 518–527 (2017).
40. Liu J, et al. Conjugate Polymer-clothed TiO₂/V₂O₅ nanobelts and their enhanced visible light photocatalytic performance in water remediation. *Journal of Colloid and Interface Science* 578, 402–411 (2020).
41. Ma R, et al. Enhanced catalytic degradation of aqueous doxycycline (DOX) in Mg-Fe-LDH@biochar composite-activated peroxymonosulfate system: Performances, degradation pathways, mechanisms and environmental implications. *Chemical Engineering Journal* 425, 131457 (2021).
42. Luo X, et al. Green synthesis of manganese-cobalt-tungsten composite oxides for degradation of doxycycline via efficient activation of peroxymonosulfate. *Journal of Hazardous Materials* 426, 127803 (2022).
43. Luo X, Asefa T, Qiu R, Su C, Cui L, Huang Z. Robust Adsorption and Persulfate-Based Degradation of Doxycycline by Oxygen Vacancy-Rich Copper-Iron Oxides Prepared through a Mechanochemical Route. *ACS ES&T Water* 2, 1031–1045 (2022).
44. Tang Y, et al. Catalytic degradation of oxytetracycline via FeVO₄ nanorods activating PMS and the insights into the performance and mechanism. *Journal of Environmental Chemical Engineering* 9, 105864 (2021).
45. Wang A, et al. MOF Derived Co-Fe nitrogen doped graphite carbon@crosslinked magnetic chitosan Micro-nanoreactor for environmental applications: Synergy enhancement effect of adsorption-PMS activation. *Applied Catalysis B: Environmental* 319, 121926 (2022).
46. Pang K, et al. Sulfur-modified chitosan derived N,S-co-doped carbon as a bifunctional material for adsorption and catalytic degradation sulfamethoxazole by persulfate. *Journal of Hazardous Materials* 424, 127270 (2022).
47. Guo R, Wang Y, Li J, Cheng X, Dionysiou DD. Sulfamethoxazole degradation by visible light assisted peroxymonosulfate process based on nanohybrid manganese dioxide incorporating ferric oxide. *Applied Catalysis B: Environmental* 278, 119297 (2020).
48. Liang J, et al. Persulfate Oxidation of Sulfamethoxazole by Magnetic Iron-Char Composites via Nonradical Pathways: Fe(IV) Versus Surface-Mediated Electron Transfer. *Environmental Science & Technology* 55, 10077–10086 (2021).
49. Tian J, et al. Efficient emerging contaminants (EM) decomposition via peroxymonosulfate (PMS) activation by Co₃O₄/carbonized polyimide (CPANI) composite: Characterization of tetracycline (TC) degradation property and application for the remediation of EM-polluted water body. *Journal of Cleaner Production* 405, 137023 (2023).
50. Wang Y-q, et al. A novel partially carbonized Fe₃O₄@PANI-p catalyst for tetracycline degradation via peroxymonosulfate activation. *Chemical Engineering Journal* 451, 138655 (2023).
51. Hu Y, et al. Singlet oxygen-dominated activation of peroxymonosulfate by passion fruit shell derived biochar for catalytic degradation of tetracycline through a non-radical oxidation pathway. *Journal of Hazardous Materials* 419, 126495 (2021).
52. Guo Y, et al. Goethite/biochar-activated peroxymonosulfate enhances tetracycline degradation: Inherent roles of radical and non-radical processes. *Science of The Total Environment* 783, 147102 (2021).

53. Jiang H-L, et al. A novel oxygen vacancies enriched CoNi LDO catalyst activated peroxymonosulfate for the efficient degradation of tetracycline. *Journal of Water Process Engineering* 52, 103526 (2023).
54. Liu L, et al. Insights into the performance, mechanism, and ecotoxicity of levofloxacin degradation in CoFe₂O₄ catalytic peroxymonosulfate process. *Journal of Environmental Chemical Engineering* 10, 107435 (2022).
55. Qin L, et al. Citrate-regulated synthesis of hydroxalcite-like compounds as peroxymonosulfate activator – Investigation of oxygen vacancies and degradation pathways by combining DFT. *Applied Catalysis B: Environmental* 317, 121704 (2022).
56. Jiang Z, et al. Electron transfer mechanism mediated nitrogen-enriched biochar encapsulated cobalt nanoparticles catalyst as an effective persulfate activator for doxycycline removal. *Journal of Cleaner Production* 384, 135641 (2023).
57. Zhao Z, Wang P, Song C, Zhang T, Zhan S, Li Y. Enhanced Interfacial Electron Transfer by Asymmetric Cu–Ov–In Sites on In₂O₃ for Efficient Peroxymonosulfate Activation. *Angewandte Chemie* 135, e202216403 (2023).
58. Tan C, et al. Activation of peroxymonosulfate by a novel EGCE@Fe₃O₄ nanocomposite: Free radical reactions and implication for the degradation of sulfadiazine. *Chemical Engineering Journal* 359, 594–603 (2019).
59. Liu C, Wang Z, Hua S, Jiao H, Chen Y, Ding D. Sewage sludge derived magnetic biochar effectively activates peroxymonosulfate for the removal of norfloxacin. *Separation and Purification Technology* 314, 123674 (2023).

Q5. Please clarify the novelty of this work compared with the recent published papers.

Response: *The novelty of this work has been reflected in various parts of the manuscript. The summary is as follows:*

In this study, we realized the transformation of the reaction mechanism of advanced oxidation process in the photo excited PMS activation system by synthesizing S-scheme heterojunction PBA/MoS₂@chitosan hydrogel.

We have conducted in-depth studies on the mechanism of efficient electron transfer at photoexcitation S-scheme heterojunction interface electric field, revealing the synergistic effect of photogenerated carrier and PMS activation,

Unlike conventional powder catalysts, the interface electric field of the S-scheme heterojunction redistributed the energy barrier and accelerated electron transport, and then changed the electronic properties and structure of transition metal activated PMS. PBA/MoS₂@CSH is cross-linked into millimeter sized hydrogel beads with 3D polymer chain closed network through natural polymer chitosan, which is conducive to rapid capture of PMS and electronic transmission in advanced oxidation process, reducing the use of transition metal activator, and limiting the leaching of metal ions by its active -NH₂ and -OH groups. Meanwhile, it can be easily separated from aqueous solutions, which provides possible convenience for practical applications. Many studies have reported the use of photocatalysts to promote Fenton like reactions, but this work proposes new explanations for this critical scientific issue, and providing insights for the future development and application of novel PMS activated functional materials in the field of water remediation.

Reviewer #2:

This manuscript reported the synthesized S-scheme heterojunction PBA/MoS₂@chitosan hydrogel to achieve the photoexcitation synergistic PMS activation. The structure characterization and analysis, discussed by the authors for the PBA/MoS₂@chitosan hydrogel catalyst, are very common. Besides, the photoexcited carriers of constructed heterojunction undergo redox conversion with PMS through an excellent S-scheme transfer pathway driven by the directional interface electric field. However, the authors proposed that the characterization and analysis, for the formed built-in electric field in the photocatalyst, are very general in photocatalytic composites, which is not better than the reported references and cannot give some valuable inspirations for rational fabrication and better understanding the photocatalytic mechanism. Overall, the characterizations are not matching with the conclusions. Therefore, this work does not have enough suitable for publication in Nat. Commun. at present level. To enhance this work, I have few minor issues and also a more general question that would need to be addressed.

Q1. The quality of English need to be improved. The abbreviation (such as ROC), subscript and superscripts should be re-checked.

Response: We thank the reviewer's comment and made the following modifications:

In Page 2 Line 17 (Abstract): "The regulation of heterogeneous material properties to enhance the degradation of emerging organic pollutants by activated peroxymonosulfate (PMS) remains a challenge." has been modified as "The regulation of heterogeneous material properties to enhance the peroxymonosulfate (PMS) activation to degrade emerging organic pollutants remains a challenge."

In Page 2 Line 22 (Abstract): "Multiple synergistic pathways greatly enhance the ability of ROS generation, leading to a significant increase in doxycycline degradation rate." has been modified as "Multiple synergistic pathways greatly enhance the reactive oxygen species generation, leading to a significant increase in doxycycline degradation rate."

In Page 2 Line 24 (Abstract): "Meanwhile, the 3D polymer chain spatial structure of chitosan hydrogel is conducive to rapid capture PMS and electron transport in advanced oxidation process." has been modified as "Meanwhile, the 3D polymer chain spatial structure of chitosan hydrogel is conducive to rapid PMS capture and electron transport in advanced oxidation process"

In Page 3 Line 31 (Introduction): "Peroxymonosulfate (PMS) can be activated to produce ROS through methods such as light irradiation" has been modified as "Peroxymonosulfate (PMS) can be activated to produce reactive oxygen species (ROS) through methods such as light irradiation"

In Page 3 Line 49 (Introduction): "Semiconductors with appropriate bandgap can generate photogenerated electrons under light irradiation" has been modified as "Semiconductors with appropriate bandgap can provide photogenerated electrons under light irradiation"

In Page 5 Line 72 (**Introduction**): “...transfer/separation interface to efficiently synergistically enhance the activation of PMS is still a **key** to be **solved**” **has been modified as** “...transfer/separation interface to efficiently synergistically enhance the activation of PMS is still a **challenge** to be **tackled**”

In Page 5 Line 76 (**Introduction**): “PBA/MoS₂@CSH **can** not only be used as a transition metal (CoFePBA) and a co-catalyst (MoS₂) to efficiently activate PMS through cycling between metal valence states. **It can**” **has been modified as** “PBA/MoS₂@CSH not only **can** be used as a transition metal (CoFePBA) and a co-catalyst (MoS₂) to efficiently activate PMS through cycling between metal valence states, **but also can**”

In Page 8 Line 134 (**Results**): “...PMS was significantly improved when the photoexcitation **condition** was **involved**. PBA/MoS₂@CSH was less...” **has been modified as** “...PMS was significantly improved when the photoexcitation was **triggered**. PBA/MoS₂@CSH was less...”

In Page 9 Line 145 (**Results**): “...DC removal ability were also significantly enhanced **in the photoexcited** compared with the catalyst...” **has been modified as** “...DC removal ability were also significantly enhanced **under photoexcitation** compared with the catalyst...”

In Page 10 Line 159 (**Results**): “The calculated energy bands of MoS₂ (**E_g** = 1.27eV) and PBA (**E_g** = 1.77eV) were basically consistent with...” **has been modified as** “The calculated energy bands of MoS₂ (**E_g** = 1.27 eV) and PBA (**E_g** = 1.77 eV) were basically consistent with...”

Fig S9 *change to*

Fig S14 *change to*

In Page 12 Line 202 (**Results**): “Fluorescence was usually produced when the excited state of the light-captured material **returns** to the ground state...” **has been modified as** “Fluorescence was usually produced when the excited state of the light-captured material **returned** to the ground state...”

In Page 12 Line 204 (Results): "...after the introduction of MoS₂ to form heterojunction, which mean that IEF interaction between catalysts..." has been modified as ".after the introduction of MoS₂ to form heterojunction, which demonstrated that IEF interaction between catalysts..."

*In Page 18 Line 310 (Results): "the solution pH was maintained between 2.7 and 3.7, which coincides with the p*H*_{IEP} range of PBA/MoS₂@CSH positive charge" has been modified as "the solution pH was maintained between 2.7 and 3.7, which coincided with the p*H*_{IEP} range of PBA/MoS₂@CSH positive charge"*

Q2. The conclusion of internal electric field is not convincing. By observing the change in interface potential from Kelvin probe force microscopy (Adv. Mater. 2021, 33, 2100317; Angew. Chem. Int. Ed. 2022, 61, e202113044) upon photoirradiation rather than just electrochemical behaviors, it is better to confirm the directional interface electric field in PBA/MoS₂@chitosan hydrogel catalyst.

Response: We thank the reviewer's comment and made the following modifications:

In Page 10 Line 176 (Results): "In order to further understand the charge transfer pathway of the IEF between MoS₂ and PBA, photoirradiated Kelvin probe force microscopy (KPFM) was applied (Figure 2d and e). The surface potential difference between point A (MoS₂) and point B (PBA) was ~35 mV before light irradiation (Figure 2f), indicating the formation of an IEF directed from point A to point B, which can serve as a driving force for photogenerated charge transfer. Upon visible light irradiation, the surface potential at point A significantly decreased, while the surface potential at point B increased. The change of interface surface potential revealed that the PBA in the heterojunction was an electron donor under illumination, as the electrons of B migrated to A, leading to an increased B potential^{38,39}."

38. Cheng C, He B, Fan J, Cheng B, Cao S, Yu J. An Inorganic/Organic S-Scheme Heterojunction H₂-Production Photocatalyst and its Charge Transfer Mechanism. *Advanced Materials* 33, 2100317 (2021).

39. Wang J, et al. Highly Durable and Fully Dispersed Cobalt Diatomic Site Catalysts for CO₂ Photoreduction to CH₄. *Angewandte Chemie International Edition* 61, e202113044 (2022).

Figure 2d, e and f has been added to the manuscript.

Fig. 2 Energy band structure and interface electric field properties. **a** The energy band and PDOS of MoS₂, PBA, and the optimized configurations of PBA/MoS₂. **b** Electrostatic potentials of MoS₂ and PBA. **c** The charge difference distribution of PBA/MoS₂ interface, red and blue indicate charge accumulation and depletion. **d, e** Atomic force microscopy image and corresponding surface potential distribution of KPFM for PBA/MoS₂@CSH in darkness and under light irradiation. **f** The line-scanning surface potential from point A to B. **g, h** The surface charge density and TRPL spectroscopy of PBA@CSH and PBA/MoS₂@CSH, respectively.

In Page 12 Line 194 (Results): “As shown in Figure 2d, the surface charge density of PBA/MoS₂@CSH...” has been modified as “As shown in Figure 2g, the surface charge density of PBA/MoS₂@CSH...”

In Page 12 Line 205 (Results): “...PBA@CSH to 3.87 ns of PBA/MoS₂@CSH (Figure 2e and Table S2) ...” has been modified as “...PBA@CSH to 3.87 ns of PBA/MoS₂@CSH (Fig. 2h and Table S3) ...”

Q3. The Mo⁴⁺ active site exposed by MoS₂ achieved high-speed electron transfer from the MoS₂ to PBA due to the work function. It accelerated the regeneration and cycling of Fe²⁺/Co²⁺ ⇌ Co³⁺/Fe³⁺, greatly promoting the activation of PMS to form free radicals. To support the S-scheme heterojunction carrier transfer mechanism, operando time-resolved spectroscopy should be considered such as the time-resolved analysis of XAS, TAS or XPS.

Response: We thank the reviewer's comment and made the following modifications:

In Page 12 Line 207 (**Results**): “However, two important issues needed to be addressed: (i) how IEF promoted charge separation; (ii) How electrons and holes were transferred at the heterojunction interface by IEF. Therefore, we further explored real-time photogenerated charge dynamics using femtosecond transient absorption spectra (TAS). The photoexcitation of PBA/MoS₂@CSH produced a ground state bleach peak (GSB) at 460 nm and a positive excited-state absorption (ESA) peak at 517 nm, which can be allocated to the electronic transition of singlet state S₁ to S_N (Fig. 3e and f)⁴², and this signal also appeared in the PBA@CSH at 588 nm (Fig. 3a and b). In addition, two stimulated emission (SE) peaks at 618 and 660 nm can be observed in PBA/MoS₂@CSH, which was the signal generated by the rapid electron–hole recombination caused by the heterojunction interface through IEF, but the SE in PBA@CSH was not significant. We also found that the ESA peak in PBA/MoS₂@CSH was more significant and another peak appeared at 700 nm, which may be attributed to the component of MoS₂ in the heterojunction. Meanwhile, it can be observed that the establishment of ESA signal for PBA@CSH required 1 ps, but PBA/MoS₂@CSH required a longer time (Fig. 3c inset), which further confirmed the strong IEF, leading to rapid recombination of electron holes. It was worth noting that the kinetic trajectory curve of PBA/MoS₂@CSH slowly rose again, as the delay time was prolonged. This was because IEF drove the electrons of PBA from the separation site to the MoS₂ interface, and the free electrons captured in the impurity state were then slowly released. As observed, PBA/MoS₂@CSH still maintained a maximum electron survival rate of 60% after a charge decay of 7500 ps⁴³. In addition, we also fitted the time distribution of TAS detected at 588 and 517 nm on PBA@CSH and PBA/MoS₂@CSH to estimate the decay kinetics of photo generated carriers (Fig. 3g). The fitting parameters were listed in Table S4, where τ₁ corresponded to the process of electron capture, and long life τ₂ was due to the interface electron transferred process. τ₃ could be attributed to the recombination of holes and impurity electrons, while A referred to the proportion of photo generated electrons involved⁴⁴. Compared to the τ₂ (20.15 ps), A₂ (0.155) and τ₃ (310.49 ps), A₃ (0.236) of PBA@CSH, the τ₂ (52.84 ps), A₂ (0.399) and τ₃ (1322.48 ps), A₃ (0.310) of PBA/MoS₂@CSH were much larger. This result indicated that IEF opened a new channel for the transfer of photogenerated electrons from PBA to MoS₂. Therefore, the τ_{avg} carrier lifetime of PBA/MoS₂@CSH had been greatly extended to 1248.32 ps, providing sufficient evidence for the enhanced photocatalytic performance of IEF, which was consistent with PL, TRPL, and photoelectrochemical tests.

42. Chen X, Wang J, Chai Y, Zhang Z, Zhu Y. Efficient Photocatalytic Overall Water Splitting Induced by the Giant

43. Li J, Cai L, Shang J, Yu Y, Zhang L. Giant Enhancement of Internal Electric Field Boosting Bulk Charge Separation for Photocatalysis. *Advanced Materials* 28, 4059–4064 (2016).
44. Xu F, et al. Step-by-Step Mechanism Insights into the TiO₂/CeS₂ S-Scheme Photocatalyst for Enhanced Aniline Production with Water as a Proton Source. *ACS Catalysis* 12, 164–172 (2022).

New Figure 3 has been added to the manuscript.

Fig. 3 Real-time photogenerated charge dynamics. a, b Transient absorption spectra of PBA@CSH and **e, f** PBA/MoS₂@CSH. **c, g** Kinetic traces as a function of probe delay time and their corresponding normalized traces after 400 nm laser pulse irradiation.

In Page 15 Line 251 (**Results**): “In situ irradiation XPS was also performed to observe the migration of photoexcited electrons. When light was applied to the PBA/MoS₂@CSH during measurement, compared to PMS in the dark, the peak of Co 2p and Fe 2p shifted positively, while the peak of Mo 3d shifted negatively with time. This result further proved the direction of photoelectron transfer from PBA to MoS₂.

In situ XPS has been added to the manuscript Fig.4a and b.

Fig. 4 Identification of S-scheme heterojunction/PMS activation active centers. **a, b** The Co 2p and Mo 3d high-resolution XPS spectra of PBA/MoS₂@CSH before and after PMS, light or Vis/PMS tests. **c** Schematic diagram of PMS activation mechanism by PBA/MoS₂@CSH transition metal and **d** PBA/MoS₂@CSH S-scheme heterojunction with interface electric field.

Q4. Reaction mechanism and steps of PMS activation by PBA/MoS₂@CSH is given, but no experimental data confirm the possible intermediates. During DC photocatalytic degradation, the in-situ diffuse reflectance infrared Fourier transform spectra measurements should be performed for detecting the reaction intermediates.

Response: We thank the reviewer's comment and made the following modifications:

In Page 18 Line 315 (**Results**): “In-situ FTIR measurements showed that the stretching vibration intensity of the S–O bond decreased significantly at 1204 and 1108 cm⁻¹ for HSO₅⁻ and SO₄²⁻ (Figure 4c). This should be the result of rapid PMS decomposition. However, no shift in the absorption band of the S–O bond of HSO₅⁻ had been observed, indicating that the O–O bond was directly broken rather than generation of metastable Mⁿ⁺–(HO)OSO₃⁻ were generated during PMS activation^{54,55}.”

54. Jin L, You S, Ren N, Ding B, Liu Y. Mo Vacancy-Mediated Activation of Peroxymonosulfate for Ultrafast Micropollutant Removal Using an Electrified MXene Filter Functionalized with Fe Single Atoms. *Environmental Science & Technology* 56, 11750-11759 (2022).

55. Dong X, et al. Diatomite supported hierarchical 2D CoNi₃O₄ nanoribbons as highly efficient

peroxymonosulfate catalyst for atrazine degradation. *Applied Catalysis B: Environmental* 272, 118971 (2020).

Figure 4c has been added to the manuscript.

Fig. 5 Reaction mechanism and steps of PMS activation by PBA/MoS₂@CSH. **a** The removal rate of DC and pH_{final} and **b** zeta potential of PBA/MoS₂@CSH at different pH. **c** In situ FTIR variation of PMS with reaction time. **d-g** The catalytic reaction route Step 1: PMS was pre-captured by protonation chitosan to form surface complex (**d**); Step 2: PMS was transferred to PBA/MoS₂ S-scheme heterojunction due to higher E_{ads} (**e**); Step 3: electron transfer and PMS activation were demonstrated by charge density difference (**f** and **g**). Yellow and green indicate charge depletion and accumulation, respectively.

In addition, we have analyzed the reaction intermediates in the DC photocatalytic degradation process through HPLC-MS and DFT. The details are as follows:

In Page 23 Line 401 (**Results**): “The intermediates and possible pathways and mechanisms for DC degradation were further investigated by HPLC-MS and DFT (Fig. 6f and S26, and Table S7). The Fukui electrophilic index (f^+) and the highest occupied molecular orbital (HOMO) of DC in the methyl region near N₂₅ indicated that it was easy to lose electrons and be attacked by electrophilic reagents (holes and ¹O₂) (Fig. S27 and Table S8). Therefore, DC may generate intermediates D1 and D5 through bond breaking or demethylation. High radical index (f^{\bullet}) indicated that the atom was easily attacked by SO₄^{•-} and [•]OH. Thus, the reaction further opened the ring and gradually broke the bond to form D2 to D6. The lowest unoccupied molecular orbital (LUMO) and nucleophilic index (f^-) in the hydroxyl functional region of O₂₀ indicated the possibility of [•]O₂⁻ attacked⁶². So it was possible to form a series of intermediate products such as D7 through functional group cleavage, and then further oxidized the final intermediate to be mineralized into H₂O and CO₂.

Fig. 5 The possible pathways for DC degradation by PBA/MoS₂@CSH under Vis/PMS.

Q5. Although the metal ions leached from the solution after degradation was lower than those directly using CoFe/MoS₂ powder, PBA/MoS₂@CSH still have a high Co, Fe and Mo contents in the degradation solution. From its practical application, it exists the secondary pollution to the environment.

Response: Our study shown that the leaching of PBA/MoS₂@CSH transition metals into the aqueous phase is not significant.

In order to more clearly express the excellent inhibition of metal precipitation of chitosan hydrogel, we have made the following specific modifications.

In Page 22 Line 376-382 (**Results**) “Therefore, the metal ions leached from the solution after degradation were much lower than those directly using CoFe/MoS₂ powder (Figure S20 and Table S3)”. **has been modified as** “Meanwhile, chitosan hydrogel entangled network space had excellent affinity for transition metals (Fig. 6d)⁶⁰. **Therefore, the metal ions leached from the solution after degradation were much lower than those directly using CoFe/MoS₂ powder and other reported catalysts (Fig. S21 and Table S5 and S6), as well as the Chinese National Standard (GB25467-2010)⁶¹.** This shown that PBA/MoS₂@CSH was very stable in the reaction, and the spherical hydrogel beads can be easily recovered after the reaction without residue, which will not cause secondary pollution to the environment.

60. Wang A, et al. MOF Derived Co-Fe nitrogen doped graphite carbon@crosslinked magnetic chitosan Micro-nanoreactor for environmental applications: Synergy enhancement effect of adsorption-PMS activation. *Applied Catalysis B: Environmental* **319**, 121926 (2022).

61. Qian J, et al. Efficient emerging contaminants (EM) decomposition via peroxymonosulfate (PMS) activation by Co₃O₄/carbonized polyaniline (CPANI) composite: Characterization of tetracycline (TC) degradation property and application for the remediation of EM-polluted water body. *Journal of Cleaner Production* **405**, 137023 (2023).

Added **Table S6** to the support information:

Table S6. Comparison with other catalysts for the Metal leaching.

Photocatalyst	Leaching concentration ($\mu\text{g L}^{-1}$)			Reference
	Co	Fe	Mo	
PBA/MoS ₂ @CSH	5.729	2.523	4.089	This work
Co/Fe/MoS ₂	23.87	18.55	19.50	Comparison
MoS ₂ /CoFe ₂ O ₄	3462	2511	/	65
Co ₃ O ₄ /Bi ₂ MoO ₆	668.0	/	/	66
sponge@MoS ₂ @GO	/	/	180.0	67
CoS@FeS	394.5	303.7	/	68
Co-CHNTs	840.0	/	/	69
CoS _x @SiO ₂	560.0	/	/	70
Co ₃ O ₄ -palygorskite	449.0	/	/	71
Co-NP	270.0	/	/	72
NiCo ₂ O ₄ NS	810.0	/	/	73
FeCo ₂ S ₄ -CN	68.00	30.00	/	74
CuS/MIL-Fe	/	510.0	/	75
Co@MoS ₂ -3	46.50	/	/	76
MoO ₃ /Bi ₂ O ₃ /g-C ₃ N ₄	/	/	~80.00	13
Fe ₃ O ₄ @CeO ₂ @BiOI	/	300.0	/	18
FeNi-LDH@biochar	/	120.0	/	77
Co ₃ O ₄ /CPANI	120.0	/	/	49
CA-LDH	354.2	/	/	55
EGCG@Fe ₃ O ₄	/	2200	/	58

65. Feng S, *et al.* MoS₂/CoFe₂O₄ heterojunction for boosting photogenerated carrier separation and the dominant role in enhancing peroxymonosulfate activation. *Chemical Engineering Journal* **433**, 134467 (2022).
66. Guo J, *et al.* Highly efficient activation of peroxymonosulfate by Co₃O₄/Bi₂MoO₆ p-n heterostructure composites for the degradation of norfloxacin under visible light irradiation. *Separation and Purification Technology* **259**, 118109 (2021).
67. Zhu L, *et al.* Designing 3D-MoS₂ Sponge as Excellent Cocatalysts in Advanced Oxidation Processes for Pollutant Control. *Angewandte Chemie International Edition* **59**, 13968–13976 (2020).
68. Wu L, *et al.* The synergy of sulfur vacancies and heterostructure on CoS@FeS nanosheets for boosting the peroxymonosulfate activation. *Chemical Engineering Journal* **446**, 136759 (2022).
69. He Z, *et al.* Amorphous cobalt oxide decorated halloysite nanotubes for efficient sulfamethoxazole degradation activated by peroxymonosulfate. *Journal of Colloid and Interface Science* **607**, 857–868 (2022).
70. Wang F, *et al.* Enhanced catalytic sulfamethoxazole degradation via peroxymonosulfate activation over amorphous CoS_x@SiO₂ nanocages derived from ZIF-67. *Journal of Hazardous Materials* **423**, 126998 (2022).
71. Yu Y, Ji Y, Lu J, Yin X, Zhou Q. Degradation of sulfamethoxazole by Co₃O₄-palygorskite composites activated peroxymonosulfate oxidation. *Chemical Engineering Journal* **406**, 126759 (2021).
72. Liu F, *et al.* Degradation of sulfamethoxazole by cobalt-nickel powder composite catalyst coupled with peroxymonosulfate: Performance, degradation pathways and mechanistic

- consideration. *Journal of Hazardous Materials* **400**, 123322 (2020).
73. Cai P, *et al.* Synergy between cobalt and nickel on NiCo₂O₄ nanosheets promotes peroxymonosulfate activation for efficient norfloxacin degradation. *Applied Catalysis B: Environmental* **306**, 121091 (2022).
 74. Li Y, *et al.* Peroxymonosulfate activation on FeCo₂S₄ modified g-C₃N₄ (FeCo₂S₄-CN): Mechanism of singlet oxygen evolution for nonradical efficient degradation of sulfamethoxazole. *Chemical Engineering Journal* **384**, 123361 (2020).
 75. Fang Z, Liu Y, Qi J, Xu Z-F, Qi T, Wang L. Establishing a high-speed electron transfer channel via CuS/MIL-Fe heterojunction catalyst for photo-Fenton degradation of acetaminophen. *Applied Catalysis B: Environmental* **320**, 121979 (2023).
 76. Li X, *et al.* Effective removal of tetracycline from water by catalytic peroxymonosulfate oxidation over Co@MoS₂: Catalytic performance and degradation mechanism. *Separation and Purification Technology* **294**, 121139 (2022).
 13. Alnaggar G, Hezam A, Drmash QA, Ananda S. Sunlight-driven activation of peroxymonosulfate by microwave synthesized ternary MoO₃/Bi₂O₃/g-C₃N₄ heterostructures for boosting tetracycline hydrochloride degradation. *Chemosphere* **272**, 129807 (2021).
 18. Kohantorabi M, Moussavi G, Oulego P, Giannakis S. Radical-based degradation of sulfamethoxazole via UVA/PMS-assisted photocatalysis, driven by magnetically separable Fe₃O₄@CeO₂@BiOI nanospheres. *Separation and Purification Technology* **267**, 118665 (2021).
 77. Mi X, Ma R, Pu X, Fu X, Geng M, Qian J. FeNi-layered double hydroxide (LDH)@biochar composite for , activation of peroxymonosulfate (PMS) towards enhanced degradation of doxycycline (DOX): Characterizations of the catalysts, catalytic performances, degradation pathways and mechanisms. *Journal of Cleaner Production* **378**, 134514 (2022)
 49. Qian J, *et al.* Efficient emerging contaminants (EM) decomposition via peroxymonosulfate (PMS) activation by Co₃O₄/carbonized polyaniline (CPANI) composite: Characterization of tetracycline (TC) degradation property and application for the remediation of EM-polluted water body. *Journal of Cleaner Production* **405**, 137023 (2023).
 55. Qin L, *et al.* Citrate-regulated synthesis of hydroxalcite-like compounds as peroxymonosulfate activator – Investigation of oxygen vacancies and degradation pathways by combining DFT. *Applied Catalysis B: Environmental* **317**, 121704 (2022).
 58. Tan C, *et al.* Activation of peroxymonosulfate by a novel EGCE@Fe₃O₄ nanocomposite: Free radical reactions and implication for the degradation of sulfadiazine. *Chemical Engineering Journal* **359**, 594–603 (2019).

Q6. Does the ratio of CoFe, MoS₂ and CSH in the heterojunction affect photocatalytic activities?

Response: We thank the reviewer's comment. As we described on page 8, line 131 "It can be observed in Fig. 1d that the efficiency of PBA/MoS₂@CSH activating PMS was significantly improved when the photoexcitation condition was involved. PBA/MoS₂@CSH was less effective (about 20%) in removing DC without the addition of PMS because they only rely on adsorption and less efficient photocatalysis."

Figure 1d. Catalytic degradation curve of catalysts.

We tested the photocatalytic degradation performance of DC with different proportions of catalysts, and the performance difference was not significant (Fig 1). However, the increase of MoS_2 content did not significantly improve the degradation of DC by PMS activation (Figure S4). The main role of photocatalysis in this study is to synergistically enhance the activation efficiency of PMS, as we described on page 16, line 280 “In PMS alone system, $\text{PBA}/\text{MoS}_2@\text{CSH}$ was used as an electron donor or acceptor to undergo redox conversion with PMS to produce ROS. However, in the Vis/PMS system, $\text{PBA}/\text{MoS}_2@\text{CSH}$ was regarded as an S-scheme heterojunction, the photoexcited electrons undergo redox conversion with PMS through an interface electric field to generate ROS, Simultaneously, photogenerated holes oxidized PMS to generate SO_5^- and $^1\text{O}_2$ or directly affect the degradation of DC. Therefore, multiple synergistic pathways greatly enhanced the ability of ROS generation and significantly increased the degradation rate of DC.”

Fig. 1. Photocatalytic degradation curve of catalysts.

Q7. When discussing the built-in electric field in the photocatalyst, it is necessary to calculate the electron band structures (CB, VB).

Response: In the manuscript, we discussed the built-in electric field through work function, theoretical calculation, Kanata model and the **new added KPFM**.

Meanwhile, the next section analyzed the carrier transfer path and calculated the electronic band structure (CB, VB). The details are as follows:

The band structure of the catalyst was studied through density functional theory (DFT) to further explain the differences in activity (Fig. S8). The calculated energy bands of MoS₂ ($E_g = 1.27$ eV) and PBA ($E_g = 1.77$ eV) were basically consistent with those in the literature and those calculated by the Kubelka–Munk (K–M) equation (where the band gaps of MoS₂ and PBA were 1.25 and 1.60 eV, respectively) (Fig. S9).

Therefore, the combination of band gap and VB–XPS directly determined the CB and VB positions of semiconductors to further explore the path and direction of carrier transport during photoexcitation (Fig. 4d and S14). The CB and VB values of PBA were -0.83 and 0.77 eV, while those of MoS₂ were -1.58 and -0.33 eV, respectively. The difference of Fermi energy levels shown in the work functions of MoS₂ and PBA calculated by DFT determined that IEF must occur at the interface, which was also consistent with the results in the previous section.

Figure S9. Kubelka–Munk energy curve plots of as-prepared catalysts.

Figure S14. VB XPS spectra and the corresponding band gap of as-prepared catalysts.

Fig. 4 c Schematic diagram of PMS activation mechanism by PBA/MoS₂@CSH transition metal and d PBA/MoS₂@CSH S-scheme heterojunction with interface electric field.

Reviewer #3:

The authors demonstrated the S-scheme heterojunction PBA/MoS₂@chitosan hydrogel, which realized the photoexcitation synergistic PMS activation system by interface electric field. It should be the first job to use S-scheme heterojunction photocatalyst to enhance PMS activation. The novel approach remarkably improved the catalytic performance, which is believed to provide good inspiration for the coupling of photocatalysis and advanced oxidation. Besides, the experiment data was logical and plentiful and stated in a systemic way, and visualization of figures was high quality. This work should be interesting to the readership of Nature Communications. However, there are some issues to be addressed, this manuscript can be accepted after minor revisions.

Q1. How is the charge transfer and hole mobility of S-scheme heterojunction? What is the difference with Type-I, Type-II, and Z-scheme?

Response: *We thank the reviewer's comment.*

Typically, there are two types of conventional heterojunction photocatalysts, those with a straddling gap (type-I), and those with a staggered gap (type-II) [1]. For the type-I heterojunction photocatalyst (see Fig. 1a), the conduction band (CB) and the valence band (VB) of semiconductor A are respectively higher and lower than the corresponding bands of semiconductor B. Therefore, under light irradiation, the electrons and holes will accumulate at the CB and the VB levels of semiconductor B, respectively. Since both electrons and holes accumulate on the same semiconductor, the electron-hole pairs cannot be effectively separated for the type-I heterojunction photocatalyst. Moreover, a redox reaction takes place on the semiconductor with the lower redox potential, thereby significantly reducing the redox ability of the heterojunction photocatalyst. For the type-II heterojunction photocatalyst (see Figure 1b), the CB and the VB levels of semiconductor A are higher than the corresponding levels of the semiconductor B. Thus, the photogenerated electrons will transfer to semiconductor B, while the photogenerated holes will migrate to semiconductor A under light irradiation, resulting in a spatial separation of electron-hole pairs. Similar to the type-I heterojunction, the redox ability of the type-II heterojunction photocatalyst will be also reduced because the reduction reaction and the oxidation reaction take place on semiconductor B with lower reduction potential and on semiconductor A with lower oxidation potential, respectively.

Figure 1 Schematic illustration of the two different types of separation of electron–hole pairs in the case of conventional light-responsive heterojunction photocatalysts: a) type-I, and b) type-II [1].

To mimic natural photosynthesis in plants and to enhance redox ability of photocatalysts, traditional Z-scheme photocatalysts were proposed by Bard in 1979, which can improve charge-separation efficiency and also reserve strong redox ability. This system is constructed of two semiconductors with suitable intermediate couples. These two semiconductors have staggered band structure configurations. In an ideal process, photogenerated holes in the VB of PC I react with electron donors (D), yielding the corresponding electron acceptors (A). Whereas the photogenerated electrons in the CB of PC II react with A, engendering D. Then, the reserved photogenerated electrons in the CB of PC I and holes in the VB of PC II participate in the reduction and oxidation reactions, respectively (Figure 2). Such a charge-transfer mode can render this system with strong redox ability, together with spatially separated redox reaction sites [2]. However, this Z-scheme heterojunction still requires another medium to act as A and D.

Figure 2. Schematic illustration of charge transfer in traditional Z-scheme heterojunction photocatalysts [2].

Based on the understanding of traditional heterojunction, a new concept of S-scheme heterojunction is proposed. This concept was first proposed by Professor Jiaguo Yu in 2019 [2][3]. Its full name is step-scheme heterojunction. This heterojunction is mainly composed of a reduced semiconductor photocatalyst (RP) with a smaller work function and a higher Fermi level and an

oxidized semiconductor photocatalyst (OP) with a larger work function and a lower Fermi level (Figure 3). It is constructed in this type manner, which can effectively realize the separation of electron-hole pairs with strong redox ability. An S-scheme heterojunction is composed of RP and OP with staggered band structures, which is similar to type-II heterojunction but with a completely different charge-transfer route. In a typical type-II heterojunction, photogenerated electrons and holes are accumulated on the CB of OP and VB of RP, respectively, resulting in weak redox ability. In an S-scheme heterojunction, the powerful photogenerated electrons and holes are reserved in the CB of RP and VB of OP, respectively, while the pointless photogenerated charge carriers are recombined, introducing a strong redox potential. In addition, it can be seen that the charge-transfer route in S-scheme mode resembles “step” in macroscopic (from low CB to high CB) and letter of N in microscopic. This is where it gets its name from.

Figure 3. (a) The work functions of g-C₃N₄ and WO₃ before contact. (b) The internal electric field and band edge bending at the interface of WO₃/g-C₃N₄ after contact. (c) The S-scheme charge transfer mechanism between WO₃ and g-C₃N₄ under light irradiation [3].

[1] J. Low, J. Yu, M. Jaroniec, S. Wageh, A.A. Al-Ghamdi, *Heterojunction Photocatalysts*, *Advanced Materials*, 29 (2017) 1601694.

[2] Q. Xu, L. Zhang, B. Cheng, J. Fan, J. Yu, *S-Scheme Heterojunction Photocatalyst*, *Chem*, 6 (2020) 1543-1559.

[3] J. Fu, Q. Xu, J. Low, C. Jiang, J. Yu, *Ultrathin 2D/2D WO₃/g-C₃N₄ step-scheme H₂-production photocatalyst*, *Applied Catalysis B: Environmental*, 243 (2019) 556-565.

Q2. Some figures have not mentioned in the manuscript, like Figure 1f, 3f, 4b and 5f (Please clarify, unless I overlooked).

Response: We thank the reviewer's comment and made the following modifications:

In Page 9 Line 151 (**Results**): "Therefore, the optical properties of the catalysts were explored through UV-vis diffuse reflectance spectra" **has been modified as** "Therefore, the optical properties of the catalysts were explored through UV-vis diffuse reflectance spectra (Fig. 1h)."

In Page 16 Line 276 (**Results**): "The results of radical quenching experiments shown that DC degradation involved a variety of pathways, including $^1\text{O}_2$ and $\cdot\text{OH}$, SO_4^- and $\cdot\text{O}_2^-$ assisted degradation." **has been modified as** "The results of radical quenching experiments shown that DC degradation involved a variety of pathways, including $^1\text{O}_2$ and $\cdot\text{OH}$, SO_4^- and $\cdot\text{O}_2^-$ assisted degradation (Fig. 4e)."

In Page 18 Line 307 (**Results**): "Interestingly, in the PBA/MoS₂@CSH synergistic PMS activation reaction, the solution pH was maintained between 2.7 and 3.7, which coincides with the p_{HIEP} range of PBA/MoS₂@CSH positive charge." **has been modified as** "Interestingly, in the PBA/MoS₂@CSH synergistic PMS activation reaction, the solution pH was maintained between 2.7 and 3.7, which coincides with the p_{HIEP} range of PBA/MoS₂@CSH positive charge (Fig. 5b)"

In Page 23 Line 401 (**Results**): "The intermediates and possible pathways and mechanisms for DC degradation were further investigated by HPLC-MS and DFT (Figure 5e and S25, and Table S4)." **has been modified as** "The intermediates and possible pathways and mechanisms for DC degradation were further investigated by HPLC-MS and DFT (Fig. 6f and S26, and Table S7)."

Q3. Many typos should be addressed such as "Eg" should be revised "Eg(subscript)", including Figure S8 and S13

Response: We thank the reviewer's comment and made the following modifications:

In Page 10 Line 159 (**Results**): "The calculated energy bands of MoS₂ ($E_g = 1.27\text{eV}$) and PBA ($E_g = 1.77\text{eV}$) were basically consistent with.." **has been modified as** "The calculated energy bands of MoS₂ ($E_g = 1.27\text{ eV}$) and PBA ($E_g = 1.77\text{ eV}$) were basically consistent with..."

Fig S9 change to

Fig S14 change to

Q4. The authors should compare the photocatalysis or PMS activation performance in this manuscript with those of reported catalytic materials.

Response: We thank the reviewer's comment and made the following modifications:

In Page 9 Line 145 (**Part 3.1**): "Compared with other 58 reported catalysts for antibiotic degradation, PBA/MoS₂@CSH had excellent catalyst-dose-rate constant k ($L \text{ min}^{-1} \text{ g}^{-1}$) at low PMS/antibiotic concentration ratio (Fig. 1g and Table S2). At the same time, the coupling system with the synergistic participation of light and PMS had a faster catalytic rate compared to the individual photocatalytic and PMS activation techniques."

Figure 1g and Table S2 has been added to the manuscript and supporting information

Fig. 1 g Comparisons of catalytic efficiency with those of reported catalysts.

Table S2. Comparison with other photocatalysts in literature.

Photocatalyst	Pollutant	Light source	PMS	Dosage	Removal Time	k (min ⁻¹)	Reference
PBA/MoS ₂ @CSH	Doxycycline (20 mg L ⁻¹)	300W XL ($\lambda \geq 420$ nm)	0.333 g L ⁻¹	0.038 g L ⁻¹	97.02% 30 min	0.1147	This study
LaFeO ₃ /SBA-15	Doxycycline (40 mg L ⁻¹)	300 W XL ($\lambda \geq 420$ nm)	0.615 g L ⁻¹	0.500 g L ⁻¹	/	0.023	2
BiFeO ₃ /SBA-15	Doxycycline (40 mg L ⁻¹)	300 W XL ($\lambda \geq 420$ nm)	1.844 g L ⁻¹	0.500 g L ⁻¹	/	0.0175	2
BiO _{1-x} Cl	Doxycycline (50 mg L ⁻¹)	5 W LED ($\lambda \geq 400$ nm)	0.250 g L ⁻¹	0.500 g L ⁻¹	79.4 105	0.0062	3
g-C ₃ N ₄ /Na-BiVO ₄	Tetracycline (20 mg L ⁻¹)	300 W XL ($\lambda \geq 420$ nm)	0.307 g L ⁻¹	0.200 g L ⁻¹	98.2% 40 min	0.109	4
Bi ₂ MoO ₆ /CuWO ₄	Tetracycline (10 mg L ⁻¹)	300 W XL ($\lambda \geq 420$ nm)	0.200 g L ⁻¹	0.200 g L ⁻¹	84.6% 20 min	0.121	5
10%Co ₃ O ₄ /g-C ₃ N ₄	Tetracycline (20 mg L ⁻¹)	300 W XL ($\lambda \geq 420$ nm)	0.061 g L ⁻¹	0.200 g L ⁻¹	98% 60 min	0.079	6
BC/CN-15	Tetracycline (10 mg L ⁻¹)	300 W XL ($\lambda \geq 420$ nm)	0.200 g L ⁻¹	0.200 g L ⁻¹	90% 60 min	0.035	7
Cu - R	Tetracycline (30 mg L ⁻¹)	300 W XL ($\lambda \geq 420$ nm)	0.300 g L ⁻¹	0.200 g L ⁻¹	96% 60 min	0.046	8
MnCo ₂ O ₄	Tetracycline (30 mg L ⁻¹)	300 W XL ($\lambda \geq 420$ nm)	0.750 g L ⁻¹	0.200 g L ⁻¹	98% 60 min	0.052	9

Photocatalyst	Pollutant	Light source	PMS	Dosage	Removal Time	k (min ⁻¹)	Reference
3NiO/g-C ₃ N ₄	Tetracycline (20 mg L ⁻¹)	300 W XL ($\lambda \geq 420$ nm)	0.100 g L ⁻¹	0.200 g L ⁻¹	/ 60 min	0.080	10
MCN	Tetracycline (10 mg L ⁻¹)	300W XL ($\lambda \geq 420$ nm)	0.500 g L ⁻¹	0.400 g L ⁻¹	89.7 30	0.061	11
MoS ₂ /Ag/g-C ₃ N ₄	Tetracycline (20 mg L ⁻¹)	300 W XL ($\lambda \geq 420$ nm)	0.061 g L ⁻¹	0.200 g L ⁻¹	91.2% 30 min	0.084	12
MoO ₃ /Bi ₂ O ₃ /g-C ₃ N ₄	Tetracycline (40 mg L ⁻¹)	Solar light	2.459 g L ⁻¹	0.600 g L ⁻¹	98% 120min	0.0248	13
CuHNPs-7.5	Tetracycline (40 mg L ⁻¹)	100W LED ($\lambda \geq 420$ nm)	0.277 g L ⁻¹	0.200 g L ⁻¹	97.8% 30 min	0.125	14
FeMo ₃ O ₃ /C ₃ N ₄ -EP	Oxytetracycline (50 mg L ⁻¹)	300 W XL ($\lambda \geq 420$ nm)	3.074 g L ⁻¹	1.000 g L ⁻¹	98.1	0.181	15
SMM-3	Levofloxacin (10 mg·L ⁻¹)	300 W XL ($\lambda \geq 420$ nm)	0.500 g L ⁻¹	0.100 g L ⁻¹	95.1% 20 min	0.196	16
Ag/AgCl@ZIF-8/g-C ₃ N ₄	Levofloxacin (10 mg·L ⁻¹)	150 W XL ($\lambda \geq 420$ nm)	1.230 g L ⁻¹	1.000 g L ⁻¹	87.3% 60 min	0.03054	17
Fe ₃ O ₄ @CeO ₂ @BiOI	Sulfamethoxazole (13 mg·L ⁻¹)	UVA-LED	0.123 g L ⁻¹	0.100 g L ⁻¹	97% 15 min	0.221	18
γ -Fe ₂ O ₃ -MnO ₂	Ciprofloxacin (17 mg·L ⁻¹)	300 W XL ($\lambda \geq 420$ nm)	0.300 g L ⁻¹	0.150 g L ⁻¹	98.3% 30 min	0.114	19
CoCr ₂ O ₄ /α-Fe ₂ O ₃ /β-La ₂ S ₃	Doxycycline (10 mg L ⁻¹)	1000W HL ($\lambda \geq 420$ nm)	/	0.050 g L ⁻¹	92.83% 345 min	0.0076	20

Photocatalyst	Pollutant	Light source	PMS	Dosage	Removal Time	k (min ⁻¹)	Reference
Co/Mn-MOF-74@g-C ₃ N ₄	Doxycycline (40 mg L ⁻¹)	300 W XL ($\lambda \geq 420$ nm)	/	0.500 g L ⁻¹	/	0.00459	21
g-C ₃ N ₄ / α -Bi ₂ (MoO ₄) ₃	Doxycycline (10 mg L ⁻¹)	500 W XL ($\lambda \geq 420$ nm)	/	1.000 g L ⁻¹	93.19% 140 min	0.0183	22
In ₂ O ₃ /Bi ₄ O ₇	Doxycycline (20 mg L ⁻¹)	300W XL ($\lambda \geq 420$ nm)	/	0.500 g L ⁻¹	92.1% 120 min	0.0197	23
AN@CN	Doxycycline (50 mg L ⁻¹)	300W XL ($\lambda \geq 420$ nm)	/	0.500 g L ⁻¹	98.67% 60 min	0.04052	24
BiM/ZnC@PANI	Doxycycline (10 mg L ⁻¹)	300W XL ($\lambda \geq 420$ nm)	/	0.100 g L ⁻¹	90% 150 min	0.0119	25
ILDAC/MIL-68(In)-NH ₂	Doxycycline (10 mg L ⁻¹)	500 W XL ($\lambda \geq 420$ nm)	/	0.200 g L ⁻¹	92% 180 min	0.00918	26
ZnO	Doxycycline (10 mg L ⁻¹)	30 W UV-C lamp	/	0.250 g L ⁻¹	~88% 780 min	0.012	27
Bi7O9I3/g-C3N4	Doxycycline (20 mg L ⁻¹)	300W XL ($\lambda \geq 420$ nm)	/	0.500 g L ⁻¹	80% 120 min	0.0125	28
BiOBr/FeWO ₄	Doxycycline (20 mg L ⁻¹)	300 W XL ($\lambda \geq 420$ nm)	/	1.000 g L ⁻¹	90.4% 60 %	0.0375	29
Nd-BiO _{2-x}	Doxycycline (10 mg L ⁻¹)	300 W XL ($\lambda \geq 420$ nm)	/	0.200 g L ⁻¹	86.14% 120 min	0.01344	30
Co ₃ O ₄ TiO ₂ /GO	Oxytetracycline (10 mg L ⁻¹)	300 W XL ($\lambda \geq 400$ nm)	/	0.250 g L ⁻¹	91% 90 min	0.0272	31

Photocatalyst	Pollutant	Light source	PMS	Dosage	Removal Time	k (min ⁻¹)	Reference
Ag/p-Ag ₂ S/n-BiVO ₄	Oxytetracycline (20 mg L ⁻¹)	500 W XL ($\lambda \geq 420$ nm)	/	0.400 g L ⁻¹	99.8% 150 min	0.0411	32
AgI/BiVO ₄	Oxytetracycline (20 mg L ⁻¹)	300 W XL ($\lambda \geq 420$ nm)	/	0.600 g L ⁻¹	80% 60 min	0.0239	33
SrTiO ₃ /BiOI	Oxytetracycline (20 mg L ⁻¹)	300 W XL ($\lambda \geq 420$ nm)	/	1.000 g L ⁻¹	85.34% 90 min	0.0252	34
Ag/N-GQDs/g-C ₃ N ₄	Tetracycline (10 mg L ⁻¹)	300 W XL ($\lambda \geq 365$ nm)	/	0.200 g L ⁻¹	92.8% 60 min	0.0428	35
CQDs/g-C ₃ N ₄	Tetracycline (10 mg L ⁻¹)	250 W XL ($\lambda \geq 420$ nm)	/	0.500 g L ⁻¹	78.6% 240 min	0.00642	36
h-BN/g-C ₃ N ₄	Tetracycline (10 mg L ⁻¹)	300 W XL ($\lambda \geq 420$ nm)	/	1.000 g L ⁻¹	79.7% 60 min	0.02775	37
h-BN/Bi ₂ MoO ₆	Tetracycline (20 mg L ⁻¹)	300W XL ($\lambda \geq 420$ nm)	/	0.500 g L ⁻¹	99.19% 140 min	0.0273	38
NGQDs-BiOI/MnNb ₂ O ₆	Tetracycline (10 mg L ⁻¹)	250 W XL ($\lambda \geq 420$ nm)	/	0.500 g L ⁻¹	87.2% 60 min	0.0331	39
TiO ₂ @V ₂ O ₅ -PPy	Tetracycline (50 mg L ⁻¹)	300W XL ($\lambda \geq 420$ nm)	/	0.600 g L ⁻¹	96% 60 min	0.04498	40
Mg-Fe LDH@bioachar	Doxycycline (35 mg L ⁻¹)	/	0.750 g L ⁻¹	0.750 g L ⁻¹	88.76% 120 min	0.23571	41
MnO/CoO/WO ₃	Doxycycline (20 mg L ⁻¹)	/	0.100 g L ⁻¹	0.500 g L ⁻¹	80.04% 120 min	0.0471	42

Photocatalyst	Pollutant	Light source	PMS	Dosage	Removal Time	k (min ⁻¹)	Reference
CuO/Fe ₂ O ₃	Doxycycline (50 mg L ⁻¹)		0.050 g L ⁻¹	0.200 g L ⁻¹	92.6% 120 min	0.04342	43
FeVO ₄ nanorods	Oxytetracycline (20 mg L ⁻¹)		0.615 g L ⁻¹	0.800 g L ⁻¹	100% 30 min	0.107	44
Co-Fe/NC ^{0.7} @GCS	Sulfamethoxazole (20 mg·L ⁻¹)		0.0615 g L ⁻¹	0.200 g L ⁻¹	90.2% 60 min	0.072	45
NSC-3	Sulfamethoxazole (20 mg·L ⁻¹)		0.307 g L ⁻¹	0.200 g L ⁻¹	98.62% 90 min	0.058	46
MF	Sulfamethoxazole (10 mg·L ⁻¹)		0.492 g L ⁻¹	0.060 g L ⁻¹	100% 80min	0.050	47
BC700Fe20	Sulfamethoxazole (10 mg·L ⁻¹)		1.230 g L ⁻¹	0.500 g L ⁻¹	82.2% 120min	0.031	48
Co ₃ O ₄ /CPANI	Tetracycline (20 mg L ⁻¹)		0.150 g L ⁻¹	0.150 g L ⁻¹	92.11% 40 min	0.09033	49
Fe ₃ O ₄ @PANI-p	Tetracycline (20 mg L ⁻¹)		2.459 g L ⁻¹	0.400 g L ⁻¹	89.8% 90 min	0.0353	50
PFSC-900	Tetracycline (20 mg L ⁻¹)		0.300 g L ⁻¹	0.400 g L ⁻¹	90.91 120	0.0317	51
Goethite/biochar	Tetracycline (30 mg L ⁻¹)		0.615 g L ⁻¹	0.050 g L ⁻¹	72.99 60	0.02062	52
Co-Ni LDO	Tetracycline (30 mg L ⁻¹)		0.984 g L ⁻¹	0.100 g L ⁻¹	100 60	~0.06	53

Photocatalyst	Pollutant	Light source	PMS	Dosage	Removal Time	k (min ⁻¹)	Reference
CoFe ₂ O ₄	Levofloxacin (5 mg L ⁻¹)		0.154 g L ⁻¹	0.100 g L ⁻¹	94.63 30	0.0997	54
CA-LDH	Ciprofloxacin (20 mg L ⁻¹)		0.500 g L ⁻¹	0.200 g L ⁻¹	98 60	0.088	55
Co@N-BC	Doxycycline (50 mg L ⁻¹)		0.307 g L ⁻¹	0.400 g L ⁻¹	92.72 30	0.0873	56
Cu-In ₂ O ₃ /O _v	Tetracycline (20 mg L ⁻¹)		0.300 g L ⁻¹	0.500 g L ⁻¹	100 20	0.2648	57
EGCG@Fe ₃ O ₄	Sulfadiazine (10 mg L ⁻¹)		0.184 g L ⁻¹	0.800 g L ⁻¹	97.9 60	0.0541	58
SBC ₈₀₀	Norfloxacin (10 mg L ⁻¹)		0.307 g L ⁻¹	0.200 g L ⁻¹	100 40	0.0785	59

- Zhao Q, Long M, Li H, Wen Q, Li D. Synthesis of MFeO₃/SBA-15 (M = La or Bi) for peroxymonosulfate activation towards enhanced photocatalytic activity. *New Journal of Chemistry* 46, 1144–1157 (2022).
- Liu M, et al. Confine activation peroxymonosulfate by surface oxygen vacancies of BiO_{1-x}Cl to boost its utilization rate. *Separation and Purification Technology* 307, 122711 (2023).
- Kang J, et al. The enhanced peroxymonosulfate-assisted photocatalytic degradation of tetracycline under visible light by g-C₃N₄/Na-BiVO₄ heterojunction catalyst and its mechanism. *Journal of Environmental Chemical Engineering* 9, 105524 (2021).
- Chen R, Dou X, Xia J, Chen Y, Shi H. Boosting peroxymonosulfate activation over Bi₂MoO₆/CuWO₄ to rapidly degrade tetracycline: Intermediates and mechanism. *Separation and Purification Technology* 296, 121345 (2022).
- Jin C, et al. Two dimensional Co₃O₄/g-C₃N₄ Z-scheme heterojunction: Mechanism insight into enhanced peroxymonosulfate-mediated visible light photocatalytic performance. *Chemical Engineering Journal* 398, 125569 (2020).
- Tang R, et al. π-π stacking derived from graphene-like biochar/g-C₃N₄ with tunable band structure for photocatalytic antibiotics degradation via peroxymonosulfate activation. *Journal of Hazardous Materials* 423, 126944 (2022).
- Yi L, Li Y, Zhu L, Gao C, Wu X. CuO decorated natural rectorite as highly efficient catalyst for photoinduced peroxymonosulfate activation towards tetracycline degradation. *Journal of Cleaner Production* 317, 128441 (2021).
- Wang J, Jiang Y, Gao C, Li Y, Wu X. Synergistic effect of bimetal in three-dimensional hierarchical MnCo₂O₄ for high efficiency of photoinduced Fenton-like reaction. *Surfaces and Interfaces* 27, 101482 (2021).
- Li S, et al. NiO/g-C₃N₄ 2D/2D heterojunction catalyst as efficient peroxymonosulfate activators toward tetracycline degradation: Characterization, performance and mechanism. *Journal of Alloys and Compounds* 880, 160547 (2021).

11. Shi H, He Y, Li Y, He T, Luo P. Efficient degradation of tetracycline in real water systems by metal-free g-C₃N₄ microspheres through visible-light catalysis and PMS activation synergy. *Separation and Purification Technology* 280, 119864 (2022).
12. Jin C, Kang J, Li Z, Wang M, Wu Z, Xie Y. Enhanced visible light photocatalytic degradation of tetracycline by MoS₂/Ag/g-C₃N₄ Z-scheme composites with peroxymonosulfate. *Applied Surface Science* 514, 146076 (2020).
13. Alnaggar G, Hezam A, Dmash QA, Ananda S. Sunlight-driven activation of peroxymonosulfate by microwave synthesized ternary MoO₃/Bi₂O₃/g-C₃N₄ heterostructures for boosting tetracycline hydrochloride degradation. *Chemosphere* 272, 129807 (2021).
14. Guo T, Jiang L, Huang H, Li Y, Wu X, Zhang G. Enhanced degradation of tetracycline in water over Cu-doped hematite nanoplates by peroxymonosulfate activation under visible light irradiation. *Journal of Hazardous Materials* 416, 125838 (2021).
15. Liu Y, et al. Enhanced activation of peroxymonosulfate by a floating FeMo₃O_x/C₃N₄ photocatalyst under visible-light assistance for oxytetracycline degradation: Performance, mechanisms and comparison with H₂O₂ activation. *Environmental Pollution* 316, 120668 (2023).
16. He Y, et al. Acceleration of levofloxacin degradation by combination of multiple free radicals via MoS₂ anchored in manganese ferrite doped perovskite activated PMS under visible light. *Chemical Engineering Journal* 431, 133933 (2022).
17. Zhou J, Liu W, Cai W. The synergistic effect of Ag/AgCl@ZIF-8 modified g-C₃N₄ composite and peroxymonosulfate for the enhanced visible-light photocatalytic degradation of levofloxacin. *Science of The Total Environment* 696, 133962 (2019).
18. Kohantorabi M, Moussavi G, Oulego P, Giannakis S. Radical-based degradation of sulfamethoxazole via UVA/PMS-assisted photocatalysis, driven by magnetically separable Fe₃O₄@CeO₂/BiOI nanospheres. *Separation and Purification Technology* 267, 118665 (2021).
19. Zhao J, Wang Y, Li N, Wang S, Yu J, Li X. Efficient degradation of ciprofloxacin by magnetic γ-Fe₂O₃-MnO₂ with oxygen vacancy in visible-light/peroxymonosulfate system. *Chemosphere* 276, 130257 (2021).
20. Sivarajani PR, et al. Fabrication of ternary nano-heterojunction via hierarchical deposition of α-Fe₂O₃ and β-La₂S₃ on cubic CoCr₂O₄ for enhanced photodegradation of doxycycline. *Journal of Industrial and Engineering Chemistry* 118, 407–417 (2023).
21. Wen Q, et al. Synergistic effect of photocatalysis and peroxymonosulfate activated by Co/Mn-MOF-74@g-C₃N₄ Z-scheme photocatalyst for removal of tetracycline hydrochloride. *Separation and Purification Technology* 313, 123518 (2023).
22. Vasanthakumar V, et al. α-Bi₂(MoO₄)₃ nanorods decorated with two-dimensional g-C₃N₄ nanosheets for efficient degradation of doxycycline under visible light illumination. *Process Safety and Environmental Protection* 163, 1–13 (2022).
23. Pan Z, Qian L, Shen J, Huang J, Guo Y, Zhang Z. Construction and application of Z-scheme heterojunction In₂O₃/Bi₄O₇ with effective removal of antibiotic under visible light. *Chemical Engineering Journal* 426, 130385 (2021).
24. Feng C, et al. Core-shell Ag₂CrO₄/N-GQDs@g-C₃N₄ composites with anti-photocorrosion performance for enhanced full-spectrum-light photocatalytic activities. *Applied Catalysis B: Environmental* 239, 525–536 (2018).
25. Wang A, et al. MOF derived ZnO clusters on ultrathin Bi₂MoO₆ yolk@shell reactor: Establishing carrier transfer channel via PANI tandem S-scheme heterojunction. *Applied Catalysis B: Environmental* 328, 122492 (2023).
26. Li D, et al. In-situ fabrication of ionic liquids/MIL-68(In)-NH₂ photocatalyst for improving visible-light photocatalytic degradation of doxycycline hydrochloride. *Chemosphere* 292, 133461 (2022).
27. Pourmoslemi S, Mohammadi A, Kobarfard F, Amini M. Photocatalytic removal of doxycycline from aqueous solution using ZnO nano-particles: a comparison between UV-C and visible light. *Water science and technology : a journal of the International Association on Water Pollution Research* 74, 1658–1670 (2016).
28. Zhang Z, Pan Z, Guo Y, Wong PK, Zhou X, Bai R. In-situ growth of all-solid Z-scheme heterojunction photocatalyst of Bi₇O₉I₃/g-C₃N₄ and high efficient degradation of antibiotic under visible light. *Applied Catalysis B: Environmental* 261, (2020).
29. Gao J, Gao Y, Sui Z, Dong Z, Wang S, Zou D. Hydrothermal synthesis of BiOBr/FeWO₄ composite photocatalysts and their photocatalytic degradation of doxycycline. *J Alloy Compd* 732, 43–51 (2018).
30. Wang Q, et al. Unsaturated Nd-Bi dual-metal sites enable efficient NIR light-driven O₂ activation for water purification. *Applied Catalysis B: Environmental* 319, 121924 (2022).
31. Jo W-K, Kumar S, Isaacs MA, Lee AF, Karthikeyan S. Cobalt promoted TiO₂/GO for the photocatalytic degradation of oxytetracycline and Congo Red. *Appl Catal B-Environ* 201, 159–

32. Wei Z, et al. A novel 3D plasmonic p-n heterojunction photocatalyst: Ag nanoparticles on flower-like p-Ag₂S/n-BiVO₄ and its excellent photocatalytic reduction and oxidation activities. *Appl Catal B-Environ* 229, 171–180 (2018).
33. Guan DL, Niu CG, Wen XJ, Guo H, Deng CH, Zeng GM. Enhanced *Escherichia coli* inactivation and oxytetracycline hydrochloride degradation by a Z-scheme silver iodide decorated bismuth vanadate nanocomposite under visible light irradiation. *J Colloid Interf Sci* 512, 272–281 (2018).
34. Wen XJ, Niu CG, Zhang L, Liang C, Zeng GM. An in depth mechanism insight of the degradation of multiple refractory pollutants via a novel SrTiO₃/BiOI heterojunction photocatalysts. *J Catal* 356, 283–299 (2017).
35. Peng Y, et al. Construction of Plasmonic Ag and Nitrogen-Doped Graphene Quantum Dots Decorated Ultrathin Graphitic Carbon Nitride Nanosheet Composites with Enhanced Photocatalytic Activity: Full-Spectrum Response Ability and Mechanism Insight. *ACS Appl Mater Inter* 9, 42816–42828 (2017).
36. Hong Y, et al. Facile fabrication of stable metal-free CQDs/g-C₃N₄ heterojunctions with efficiently enhanced visible-light photocatalytic activity. *Sep Purif Technol* 171, 229–237 (2016).
37. Jiang L, et al. Metal-free efficient photocatalyst for stable visible-light photocatalytic degradation of refractory pollutant. *Appl Catal B-Environ* 221, 715–725 (2018).
38. Du Z, et al. Ultrathin h-BN/Bi₂MoO₆ heterojunction with synergetic effect for visible-light photocatalytic tetracycline degradation. *J Colloid Interface Sci* 589, 545–555 (2021).
39. Yan M, et al. Fabrication of nitrogen doped graphene quantum dots-BiOI/MnNb₂O₆ p-n junction photocatalysts with enhanced visible light efficiency in photocatalytic degradation of antibiotics. *Appl Catal B-Environ* 202, 518–527 (2017).
40. Liu J, et al. Conjugate Polymer-clothed TiO₂/V₂O₅ nanobelts and their enhanced visible light photocatalytic performance in water remediation. *Journal of Colloid and Interface Science* 578, 402–411 (2020).
41. Ma R, et al. Enhanced catalytic degradation of aqueous doxycycline (DOX) in Mg-Fe-LDH@biochar composite-activated peroxymonosulfate system: Performances, degradation pathways, mechanisms and environmental implications. *Chemical Engineering Journal* 425, 131457 (2021).
42. Luo X, et al. Green synthesis of manganese-cobalt-tungsten composite oxides for degradation of doxycycline via efficient activation of peroxymonosulfate. *Journal of Hazardous Materials* 426, 127803 (2022).
43. Luo X, Asefa T, Qiu R, Su C, Cui L, Huang Z. Robust Adsorption and Persulfate-Based Degradation of Doxycycline by Oxygen Vacancy-Rich Copper-Iron Oxides Prepared through a Mechanochemical Route. *ACS ES&T Water* 2, 1031–1045 (2022).
44. Tang Y, et al. Catalytic degradation of oxytetracycline via FeVO₄ nanorods activating PMS and the insights into the performance and mechanism. *Journal of Environmental Chemical Engineering* 9, 105864 (2021).
45. Wang A, et al. MOF Derived Co-Fe nitrogen doped graphite carbon@crosslinked magnetic chitosan Micro-nanoreactor for environmental applications: Synergy enhancement effect of adsorption-PMS activation. *Applied Catalysis B: Environmental* 319, 121926 (2022).
46. Pang K, et al. Sulfur-modified chitosan derived N,S-co-doped carbon as a bifunctional material for adsorption and catalytic degradation sulfamethoxazole by persulfate. *Journal of Hazardous Materials* 424, 127270 (2022).
47. Guo R, Wang Y, Li J, Cheng X, Dionysiou DD. Sulfamethoxazole degradation by visible light assisted peroxymonosulfate process based on nanohybrid manganese dioxide incorporating ferric oxide. *Applied Catalysis B: Environmental* 278, 119297 (2020).
48. Liang J, et al. Persulfate Oxidation of Sulfamethoxazole by Magnetic Iron-Char Composites via Nonradical Pathways: Fe(IV) Versus Surface-Mediated Electron Transfer. *Environmental Science & Technology* 55, 10077–10086 (2021).
49. Tian J, et al. Efficient emerging contaminants (EM) decomposition via peroxymonosulfate (PMS) activation by Co₃O₄/carbonized polyimide (CPANI) composite: Characterization of tetracycline (TC) degradation property and application for the remediation of EM-polluted water body. *Journal of Cleaner Production* 405, 137023 (2023).
50. Wang Y-q, et al. A novel partially carbonized Fe₃O₄@PANI-p catalyst for tetracycline degradation via peroxymonosulfate activation. *Chemical Engineering Journal* 451, 138655 (2023).
51. Hu Y, et al. Singlet oxygen-dominated activation of peroxymonosulfate by passion fruit shell derived biochar for catalytic degradation of tetracycline through a non-radical oxidation pathway. *Journal of Hazardous Materials* 419, 126495 (2021).
52. Guo Y, et al. Goethite/biochar-activated peroxymonosulfate enhances tetracycline degradation: Inherent roles of radical and non-radical processes. *Science of The Total Environment* 783, 147102 (2021).

53. Jiang H-L, et al. A novel oxygen vacancies enriched CoNi LDO catalyst activated peroxymonosulfate for the efficient degradation of tetracycline. *Journal of Water Process Engineering* 52, 103526 (2023).
54. Liu L, et al. Insights into the performance, mechanism, and ecotoxicity of levofloxacin degradation in CoFe₂O₄ catalytic peroxymonosulfate process. *Journal of Environmental Chemical Engineering* 10, 107435 (2022).
55. Qin L, et al. Citrate-regulated synthesis of hydroxalcite-like compounds as peroxymonosulfate activator – Investigation of oxygen vacancies and degradation pathways by combining DFT. *Applied Catalysis B: Environmental* 317, 121704 (2022).
56. Jiang Z, et al. Electron transfer mechanism mediated nitrogen-enriched biochar encapsulated cobalt nanoparticles catalyst as an effective persulfate activator for doxycycline removal. *Journal of Cleaner Production* 384, 135641 (2023).
57. Zhao Z, Wang P, Song C, Zhang T, Zhan S, Li Y. Enhanced Interfacial Electron Transfer by Asymmetric Cu–Ov–In Sites on In₂O₃ for Efficient Peroxymonosulfate Activation. *Angewandte Chemie* 135, e202216403 (2023).
58. Tan C, et al. Activation of peroxymonosulfate by a novel EGCE@Fe₃O₄ nanocomposite: Free radical reactions and implication for the degradation of sulfadiazine. *Chemical Engineering Journal* 359, 594–603 (2019).
59. Liu C, Wang Z, Hua S, Jiao H, Chen Y, Ding D. Sewage sludge derived magnetic biochar effectively activates peroxymonosulfate for the removal of norfloxacin. *Separation and Purification Technology* 314, 123674 (2023).

Q5. Some statements in results and discussion should be supported with relevant references, such as: Page 7 “The C≡N vibration peak ... Because there were different coordination modes in the PBA structure” Page 20 “The Fukui electrophilic index.... indicated that it was easy to lose electrons and be attacked by electrophilic reagents”...

Response: We thank the reviewer's comment and made the following modifications:

In Page 7 Line 118 (**Results**): “Because there were different coordination modes in the PBA structure, it had a certain number of open channels formed by unbounded nitroso and coordinated water molecules (Fig. S6b inset)¹⁰.”

10. Meng X, et al. Light-Driven CO₂ Reduction over Prussian Blue Analogues as Heterogeneous Catalysts. *ACS Catalysis* 12, 89-100 (2022).

In Page 23 Line 403 (**Results**): “The Fukui electrophilic index (f^+) and the highest occupied molecular orbital (HOMO) of DC in the methyl region near N₂₅ indicated that it was easy to lose electrons and be attacked by electrophilic reagents (holes and ¹O₂) (Fig. S27 and Table S8)²¹.”

21. Wang A, et al. MOF derived ZnO clusters on ultrathin Bi₂MoO₆ yolk@shell reactor: Establishing carrier transfer channel via PANI tandem S-scheme heterojunction. *Applied Catalysis B: Environmental* 328, 122492 (2023).

Q6. Some minor English linguistic errors and formatting issues are present. Double-check the grammar for the whole manuscript.

Response: We thank the reviewer's comment and made the following modifications:

In Page 2 Line 17 (**Abstract**): “The regulation of heterogeneous material properties to enhance the degradation of emerging organic pollutants by activated peroxymonosulfate (PMS) remains a challenge.” **has been modified as** “The regulation of heterogeneous material properties to enhance the peroxymonosulfate (PMS) activation to degrade emerging organic pollutants remains a challenge.”

In Page 2 Line 22 (**Abstract**): “Multiple synergistic pathways greatly enhance the ability of ROS generation, leading to a significant increase in doxycycline degradation rate.” **has been modified as** “Multiple synergistic pathways greatly enhance the reactive oxygen species generation, leading to a significant increase in doxycycline degradation rate.”

In Page 2 Line 24 (**Abstract**): “Meanwhile, the 3D polymer chain spatial structure of chitosan hydrogel is conducive to rapid capture PMS and electron transport in advanced oxidation process.” **has been modified as** “Meanwhile, the 3D polymer chain spatial structure of chitosan hydrogel is conducive to rapid PMS capture and electron transport in advanced oxidation process”

In Page 2 Line 26 (**Abstract**): “There is reason to believe that the synergistic activation of PMS by S–scheme **heterojunctions** regulated...” **has been modified as** “There is reason to believe that the synergistic activation of PMS by S–scheme **heterojunction** regulated”

In Page 3 Line 31 (**Introduction**): “Peroxymonosulfate (PMS) can be activated to produce **ROS** through methods such as light irradiation” **has been modified as** “Peroxymonosulfate (PMS) can be activated to produce **reactive oxygen species (ROS)** through methods such as light irradiation”

In Page 3 Line 49 (**Introduction**): “Semiconductors with appropriate bandgap can **generate** photogenerated electrons under light irradiation” **has been modified as** “Semiconductors with appropriate bandgap can **provide** photogenerated electrons under light irradiation”

In Page 4 Line 66 (**Introduction**): “Among the candidates for constructing **heterojunctions**, molybdenum disulfide (MoS_2), a layered transition metal chalcogenide, exhibits great potential in PMS activation and catalysis” **has been modified as** “Among the candidates for constructing **heterojunction**, molybdenum disulfide (MoS_2), a layered transition metal chalcogenide, exhibits great potential in PMS activation and catalysis”

In Page 5 Line 72 (**Introduction**): “...transfer/separation interface to efficiently synergistically enhance the activation of PMS is still a **key** to be **solved**” **has been modified as** “...transfer/separation interface to efficiently synergistically enhance the activation of PMS is still a **challenge** to be **tackled**”

In Page 5 Line 76 (**Introduction**): “PBA/ MoS_2 @CSH **can** not only be used as a transition metal (CoFePBA) and a co–catalyst (MoS_2) to efficiently activate PMS through cycling between metal valence states. **It can**” **has been modified as** “PBA/ MoS_2 @CSH not only **can** be used as a transition metal (CoFePBA) and a co–catalyst (MoS_2) to efficiently activate PMS through cycling between metal valence states, **but also can**”

In Page 8 Line 134 (**Results**): “...PMS was significantly improved when the photoexcitation **condition** was **involved**. PBA/ MoS_2 @CSH was less...” **has been modified as** “...PMS was significantly improved when the photoexcitation was **triggered**. PBA/ MoS_2 @CSH was less...”

In Page 9 Line 145 (**Results**): “...DC removal ability were also significantly enhanced **in the photoexcited** compared with the catalyst...” **has been modified as** “...DC removal ability were also significantly enhanced **under photoexcitation** compared with the catalyst...”

In Page 10 Line 159 (**Results**): “The calculated energy bands of MoS_2 ($E_g = 1.27\text{eV}$) and PBA ($E_g = 1.77\text{eV}$) were basically consistent with..” **has been modified as** “The calculated energy bands of MoS_2 ($E_g = 1.27\text{ eV}$) and PBA ($E_g = 1.77\text{ eV}$) were basically consistent with...”

In Page 10 Line 162 (**Results**): “The total density of states (DOS) calculated in Figure 2a shown that the top of the **valence band (VB)** of PBA/ MoS_2 ” **has been modified as** “The total density of

states (DOS) calculated in Fig. 2a shown that the top of the VB of PBA/MoS₂

In Page 10 Line 165 (Results): “while the bottom of the conduction band (CB) was mainly composed of Mo and S.” **has been modified as** “while the bottom of the CB was mainly composed of Mo and S.”

In Page 12 Line 202 (Results): “Fluorescence was usually produced when the excited state of the light-captured material returns to the ground state...” **has been modified as** “Fluorescence was usually produced when the excited state of the light-captured material returned to the ground state..”

In Page 12 Line 204 (Results): “...after the introduction of MoS₂ to form heterojunction, which mean that IEF interaction between catalysts...” **has been modified as** “...after the introduction of MoS₂ to form heterojunction, which demonstrated that IEF interaction between catalysts...”

In Page 16 Line 274 (Results): “Therefore, S-scheme heterojunctions can vividly describe the photogenerated carrier transfer path and confer higher redox capacity on PBA/MoS₂@CSH” **has been modified as** “Therefore, S-scheme heterojunction can vividly describe the photogenerated carrier transfer path and confer higher redox capacity on PBA/MoS₂@CSH”

In Page 16 Line 274 (Results): “This provides strong evidence for the synergistic driving of PMS activation by photoexcitation S-scheme heterojunction with IEF to enhance the ROS production.” **has been modified as** “This provided strong evidence for the synergistic driving of PMS activation by photoexcitation S-scheme heterojunction with IEF to enhance the ROS production.”

In Page 18 Line 300 (Results): “...and S-scheme heterojunctions was excited by light to generate photogenerated carriers...” **has been modified as** “and S-scheme heterojunction was excited by light to generate photogenerated carriers,”

In Page 18 Line 310 (Results): “the solution pH was maintained between 2.7 and 3.7, which coincides with the pH_{IEP} range of PBA/MoS₂@CSH positive charge” **has been modified as** “the solution pH was maintained between 2.7 and 3.7, which coincided with the pH_{IEP} range of PBA/MoS₂@CSH positive charge”

In Page 19 Line 331 (Results): “The activation of PMS and the use of electron dense regions generated by photoexcited heterojunctions to induce the DC degradation through free/non-free radicals involved electron migration” **has been modified as** “The activation of PMS and the use of electron dense regions generated by photoexcited heterojunction to induce the DC degradation through free/non-free radicals involved electron migration”

In Page 21 Line 364 (Part 3.5): “It is well known that temperature rise was conducive to molecular diffusion and PMS decomposition (endothermic reaction)” **has been modified as** “It was well known that temperature rise was conducive to molecular diffusion and PMS decomposition (endothermic reaction)”

REVIEWERS' COMMENTS

Reviewer #1 (Remarks to the Author):

I have reviewed the revised manuscript, and the author has made relevant changes based on my comments. I would recommend the publication at this stage.

Reviewer #2 (Remarks to the Author):

Revision is reasonably done and this version manuscript can be acceptable for publication.

Reviewer #3 (Remarks to the Author):

I reviewed the revised manuscript very carefully and all of my comments are implemented. Thus, the revised manuscript can be accepted now.

Response to Reviews

Reviewer #1:

I have reviewed the revised manuscript, and the author has made relevant changes based on my comments. I would recommend the publication at this stage.

Response: We thank the reviewer's comment and efforts on our work.

Reviewer #2:

Revision is reasonably done and this version manuscript can be acceptable for publication.

Response: We thank the reviewer's comment and efforts on our work.

Reviewer #3:

I reviewed the revised manuscript very carefully and all of my comments are implemented. Thus, the revised manuscript can be accepted now.

Response: We thank the reviewer's comment and efforts on our work.